# Estimation of regional meteorological aridity and drought characteristics in Baluchistan province, Pakistan

**Muhammad Rafiq**[1◉], **Yue Cong Li**[1]*◉, **Yanpei Cheng**[2◉], **Ghani Rahman**[3]*, **Yuanjie Zhao**[4‡], **Hammed Ullah Khan**[5‡]

**1** Key Laboratory of Environmental Evolution and Ecological Construction of Hebei Province, Hebei Normal University, 050024, Shijiazhuang, PR China, **2** Departments of Information and Mapping Institute of Hydrogeology and Environmental Geology, CAGS, Beijing, China, **3** Department of Geography University of Gujrat, Punjab, Pakistan, **4** Department of Environmental and Resource Sciences, Hebei Normal University China, Shijiazhuang, China, **5** Geological Survey of Pakistan, Quetta, Pakistan

◉ These authors contributed equally to this work.
‡ These authors also contributed equally to this work.
* lyczhli@aliyun.com (YCL); ghani.rahman@uog.edu.pk (GR)

**Data Availability Statement:** The authors received the from Pakistan Meteorological Department upon request. The authors did not receive any special privileges in accessing the data. Individual

## Abstract

Droughts and prevailing arid conditions have a significant impacts on the natural environment, agriculture, and human life. To analyze the regional characteristics of drought in Baluchistan province, the aridity index (AI) and standardized potential evapotranspiration index (SPEI) were used in. The study analyzed the rainfall, temperature, and potential evapotranspiration (PET) data and the same were used for the calculation of AI as well as SPEI to find out the drought spells during the study period. The linear regression and Mann-Kendall test were applied to calculate the trend in AI as well as in SPEI results. The AI results revealed that most of the meteorological stations are arid and semi-arid, where the highest increasing aridity is noted at Kalat (0.0065/year). The results of the SPEI at 1 and 6-months identified the extreme to severe drought spell during 1998–2002 in all meteorological stations of Baluchistan province. The distinct drought spells identified from the SPEI results were in the years 1998–2003, 2006–2010, 2015–2016 and 2019. The drought frequency results showed highest frequency percentage at Lasbella (46%) of extreme to severe drought. The Mann-Kendall trend results showed negative trend in monthly AI and 1-month SPEI results and most significant trend was observed in April and October months, this shows that aridity and drought in the region are decreasing to some extent except Dalbandin and Lasbella observed increasing trend in winter season (November to January months) and Kalat met-station observed increasing trend in June. Prior investigation and planning of drought situations can help in controlling the far-reaching consequences on environment and human society.

researchers can contact them through the following Email and Contact numbers to obtain the required Data from Pakistan Meteorological Department. Address: Climate Data Processing Centre (CDPC), KHI Phone: +92-21-99261413; +92-21-99261438 Email: info.cdpc@pmd.gov.pk Website: http://www.pmd.gov.pk/cdpc/home.htm.

**Funding:** This study is supported by National Natural Science Foundation of China (Grant No. 41877433 and U20A20116) The funders had no role in study design, data collection and analysis, decision to publish, or preparation of the manuscript.

**Competing interests:** The authors have declared that no competing interests exist.

## Introduction

There has been a noticeable warming tendency in the global climate over the past century [1,2]. The combined effects of climate change and human activity, around the world have increased the frequency and intensity of extreme temperature and intense precipitation. The 20th century witnessed a substantial increase in anthropogenic activities that leaded significant changes in the natural environment that ultimately resulted in rise of temperatures [3]. It is expected that climate change will disrupt the global hydrological cycle by altering the natural precipitation patterns [4,5] that will results water scarcity, aridity, desertification, droughts, and floods [6]. The aridity is a persistent shortage of moisture for long period while shortage in precipitation in an area is known as drought [7]. Drought is an environmental hazard that have negative impacts on natural environment and human social structures [8]. The severe effects of droughts are the disruption of the social structure as it compels the people of the affected on migration, cause agricultural losses, and affect the water-dependent activities [9]. According Ofipcc (2013), drought is the time of unusually dry weather conditions long enough to create severe hydrological imbalance. In general, droughts occur in all climate zones, and they are defined as an area receiving little precipitation than normal. It is caused by a number of natural phenomena including high temperature, strong winds, and less rainfall [5,10].

The droughts are classified in four major types: meteorological, agricultural, hydrological, and socioeconomic [11]. All types of droughts commenced with the shortage of precipitation in a particular region and that is termed as meteorological drought [12,13] which leads to reduction in stream flow, lowering of the groundwater table, and decrease in the level of soil moisture are all the indicators of hydrological drought [14]. The drought affects agriculture as it increase stress and reduce the crops production [15].

The Asian countries including India, Pakistan, Afghanistan, and Sri Lanka have recorded drought in average every three years during the past fifty years [16]. Droughts in Pakistan are common as most of the country experiencing dry climate, with only small area in the north experiencing humid conditions [17–19]. Most of the Baluchistan, Sindh and the southern and central parts of Punjab receives less than 250 mm of rainfall annually [17]. Agriculture is the backbone of the country economy Rahman et al. [20] that mostly depends on rainfall and irrigation from the rivers feed by glaciers present in the northern areas of Pakistan and Kashmir [21]. The country experiencing significant spatio-temporal variations in rainfall and snow cover areas which affect the agriculture in terms of yield and production [8,22,23]. According to climate projections, the temperature in Pakistan will rise 1.1˚C to 6.4˚C in 2100 [24]. It will increase the threat of drought particularly in the arid regions of Pakistan, like the Baluchistan Province [25]. The province has already experienced serious droughts in 1967–1969, 1971, 1973–75, 1994, and 1998–2002, and all of which have caused severe impacts on the economy and people of the province [26,27]. The extended 1998–2002 drought affected 80% of the crop plants, cause decrease in crop production and death of two million livestock [28,29]. Beside this, the frequent droughts and excessive water demand have caused substantial decline in the water table [30].

A number of studies have been conducted on drought in Pakistan [18,30–37] and some are specifically on Baluchistan [25–27,29,38–43]. Most of these studies focused on precipitation variation and meteorological drought in Pakistan and Baluchistan. In meteorological drought the indices mainly depends on the variation in temperature while they ignore the role of temperature and rate of evapotranspiration. The climate change experts predicts the increase in global temperature which ultimately leads to rise the rate of evapotranspiration and drought [19,36,44]. Therefore, SPEI drought index has been applied in this study which incorporate both the effect of temperature change taking into account the rate of evapotranspiration and

precipitation [45]. The main aim of this research is to determine the intensity of drought and variations in precipitation in various areas of Baluchistan province using the Standardized Precipitation Evapotranspiration Index (SPEI) and Aridity Index. The linear regression is applied in this study to find out the spatio-temporal trend of aridity at different met-stations. The Mann-Kendall trend (MK) test is applied to find out the significant trends in monthly $A_I$ and SPEI.

## The study area

The physiography of Baluchistan province mostly covered by mountains (Sulaiman Kirthar ranges), plains with arid and semi-arid climatic conditions, Baluchistan plateau and Kharan desert in southwest of the study area (Fig 1). The Sulaiman-Kirthar mountains extend from the northeast to the south-east up to Makran ranges in coastal areas having rugged and dry topography. It is the largest province of Pakistan in area, spanning over 347,220 square kilometers which is 44% of the country area [46]. Baluchistan is located 24.89˚ to 32.098˚ N latitude and 60.87˚ to 70.30˚ E longitude. Afghanistan lies to the west, Iran to the southwest, Arabian

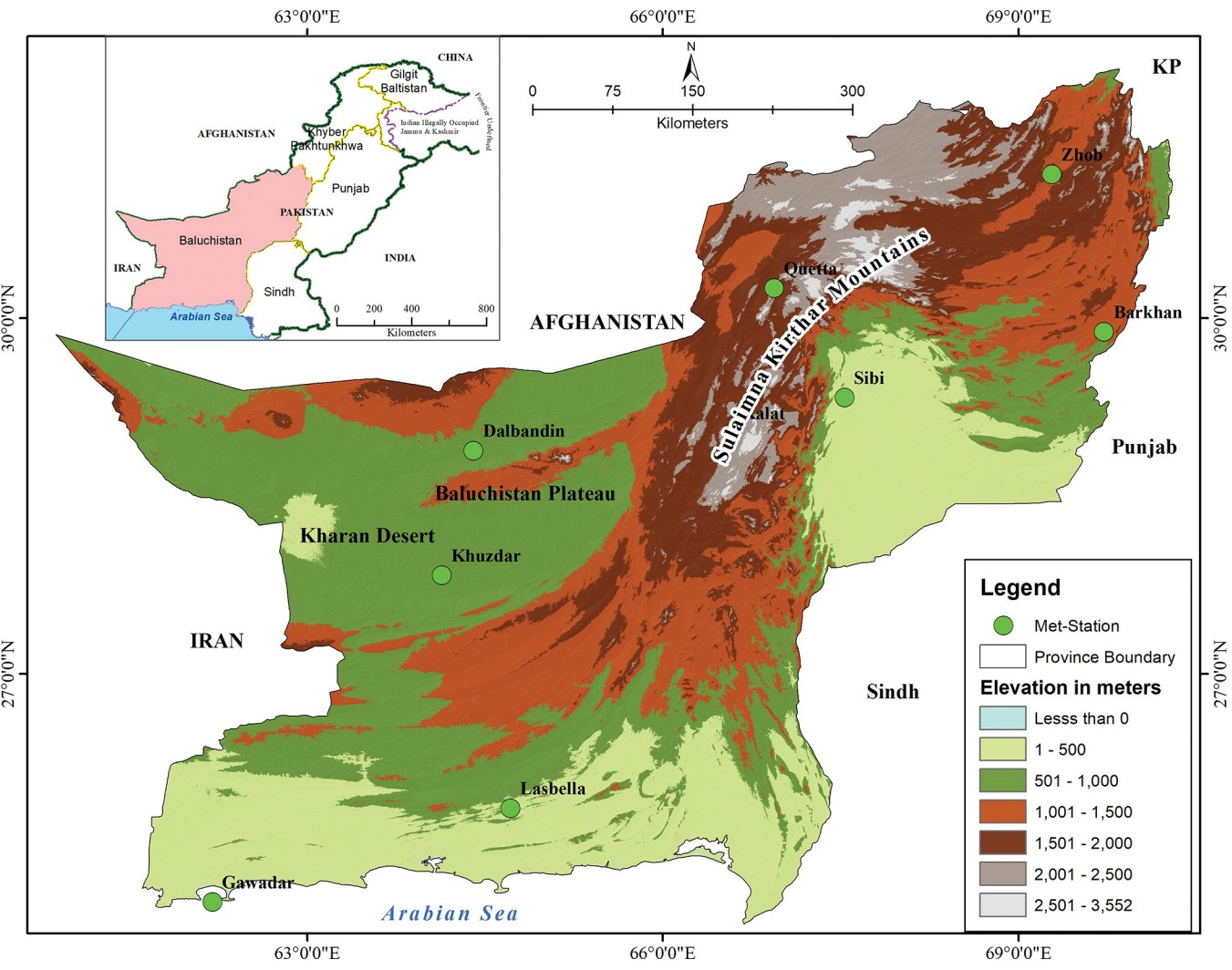

**Fig 1. The location and physiographic map of the Baluchistan.**

**Table 1. The location and elevation information of the meteorological stations.**

| Met-station | Time Period | Latitude | Longitude | Elevation (in meters) |
|---|---|---|---|---|
| Dalbandin | 1986–2021 | 28.8830˚ | 64.4000˚ | 850.09 |
| Kalat | 1986–2021 | 29.0330˚ | 66.5830˚ | 2016.86 |
| Lasbella | 1986–2021 | 25.8700˚ | 64.7129˚ | 53.95 |
| Quetta | 1986–2021 | 30.2510˚ | 66.9380˚ | 1600.20 |
| Khuzdar | 1986–2021 | 27.8330˚ | 64.1330˚ | 1232.00 |
| Zhob | 1980–2021 | 31.21˚ | 69.28˚ | 1561.49 |
| Sibi | 1980–2021 | 29.33˚ | 67.53˚ | 436.35 |

Sea in south, Khyber Pakhtunkhwa lies to north of Baluchistan, Sindh and Punjab to the east of the province [47]. Baluchistan plateau is to the west of the Sulaiman Kirthar ranges while Kharan desert is in the southwest of the province [48]. The province is important for trade with neighboring Islamic countries and Arab states. In the near future, it will serve as a gateway and a center for regional as well as global trade due to the development of Gawadar port [49]. The 44% of the country total area (Baluchistan province) only having 5% (8 million) of the country population [38]. The province receives rainfall from monsoon in summer and western disturbances in winter season. Most of the Baluchistan is semi-arid to hyper-arid where the annual rainfall ranges from 30 mm to 397 mm [24,50].

## Research methodology

### Data collection and method

To achieve the objectives of the study, the temperature and rainfall data was collected from the Pakistan meteorological department for different meteorological stations (Table 1). As all the meteorological stations are not established in the same time therefore there are variation in the data i.e. data of Dalbandin, Kalat, Lasbella, Quetta, and Khuzdar was available from 1986 to 2021 while Sibi and Zhob was from 1980–2021 (Table 1). The data was available in monthly averages for temperature and monthly sum of rainfall. The temperature and rainfall data was further processed for the calculation of SPEI, potential evapotranspiration and Aridity Index. The SPEI was calculated for 1 and 6-month time scale using R studio. The linear regression, standard deviation and Mann-Kendall trend test were applied for further quantifying trend in the Aridity Index and SPEI results.

### Aridity index

The aridity index ($A_I$) quantify the level of dryness at a particular area or meteorological station based on precipitation and potential evapotranspiration (Table 2). The aridity index was

**Table 2. The climate classification based Aridity Index values [52].**

| Aridity Index Value | Climate Class |
|---|---|
| < 0.03 | Hyper Arid |
| 0.03–0.2 | Arid |
| 0.2–0.5 | Semi-Arid |
| 0.5–0.65 | Dry sub-humid |
| > 0.65 | Humid |

calculated using the following equation in R studio [51]:

$$A_I = P/PET \tag{1}$$

P represents the annual precipitation and PET is annual potential evapotranspiration. The PET was calculated using the Thornthwaite method.

## Standardized Precipitation Evapotranspiration Index (SPEI)

SPEI calculate the water balance between precipitation and potential evapotranspiration (PET) to calculate drought and wet conditions in a particular area [45]. The SPEI consider the effect of temperature and precipitation on drought and thus combines the effect of both precipitation and temperature on drought. Most of the met-stations in the world do not directly calculate PET therefore numerous techniques have been developed to calculate it indirectly from accessible meteorological factors [53]. Some example of the famous methodologies for PET calculation are Penman Monteith, the Hargreaves approach [54] and the Thornthwaite method [55]. In this study, the Thornthwaite technique was used to compute PET. The drought are characterized in different classes based on SPEI values as shown in Table 3.

The following formula is used to compute PET:

$$\text{PET} = 16 \text{ x } \left(\frac{N}{12}\right) \times \left(\frac{m}{30}\right) \text{ x } \left(10 \text{ x } \frac{T_i}{I}\right)^{\alpha} \tag{2}$$

N signifies the average monthly sunlight hours, m the number of days in a month, Ti the monthly average temperature in Celsius, and α is the coefficient dependent on I [45].

$$\alpha = 6.75 \text{ x } 10^{-7} \text{ x } I^3 - 7.71 \text{ x } 10^{-5} \text{ x } I^2 + 1.79 \text{ x } 10^{-2} \text{ x } I + 0.49 \tag{3}$$

In this equation, I use the following expression to express the thermal index obtained from the total of the 12-month index values:

$$I = \sum_{i=1}^{12} \text{ x } \left(\frac{T_i}{5}\right)^{1.514} \tag{4}$$

The difference between precipitation and PET (water balance) was calculated using the following formula after obtaining the PET [55].

$$D_i = P_i - PET_i \tag{5}$$

The Di is the water balance calculated from the difference between precipitation and PET, which shows whether there is a water surplus or deficit for the month under consideration. Using the following equation, the Di results are accumulated over various time frames.

$$D_n^k = \sum_{i-1}^{k-1} (p_{n-1} - PET_{n-1}), \text{ n} \geq \text{k} \tag{6}$$

**Table 3. Drought intensity based on SPEI value.**

| SPI value | Category |
|---|---|
| $\geq 2$ | Extreme wet |
| 1.5 to 1.99 | Severe wet |
| 1.0 to 1.49 | Moderate wet |
| $-0.99$ to 0.99 | Near normal |
| $-1.0$ to $-1.49$ | Moderately drought condition |
| $-1.5$ to $-1.99$ | Severely drought condition |
| $\leq -2.0$ | Extremely drought condition |

The time scale of the data is expressed by k, and the computation frequency is expressed by n in this equation. SPEI requires three distribution parameters: Pearson III, lognormal, and extreme values, which distinguishes it from SPI, which only requires two [45]. In the two-parameter distribution, the variable x has a lower value limit of zero (0>x1), however in the three-parameter distribution, x can take values in the range (>x1), where is the distribution's origin parameter. The variable x can take on negative values, and negative values are prevalent in the D series. The log-logistic probability density function was used to represent the Di values using the three parameters.

$$f(x) = \frac{\beta}{\alpha}\left(\frac{x - \Upsilon}{\alpha}\right)^{\beta-1}\left[1 + \frac{x - \Upsilon}{\alpha}\right]^{-2} \tag{7}$$

The α scale parameter, the β shape parameter, and the γ origin parameter are all obtained using the L-moment procedure via the equations below:

$$\alpha = \frac{(W_0 - 2W_1)\beta}{\Gamma\left(1 + \frac{1}{\beta}\right)\Gamma\left(1 - \frac{1}{\beta}\right)} \tag{8}$$

$$\beta = \frac{(2W_1 - W_0)}{6W_1 - W_0 - 6W_2} \tag{9}$$

$$\Upsilon = W_0 - \alpha\,\Gamma\left(1 + \frac{1}{\beta}\right)\Gamma\left(1 - \frac{1}{\beta}\right) \tag{10}$$

where Γ(β) is the gamma function of β. The probability distribution function of the log-logistic distribution for D series data is given by the following expression:

$$F(X) = \left[\left(1 + \frac{\alpha}{x - Y}\right)^{\beta}\right]^{-1} \tag{11}$$

The SPEI may be simply determined using F(x) as the standardized values of F(x) by using the following equation:

$$\text{SPEI} = W - \frac{C0 + C1W + C2Wd + bc}{1 + d1w + d2w2 + d3w3} \tag{12}$$

Where W = $\sqrt{-2ln(p)}$, for P ≤ 0.5 and P is the probability of exceeding a determined D value, P = 1- F (X); when P > 0.5 in the above equations the given below values are constant

C0 = 2.515515 C1 = 0.802853 C2 = 0.010328 d1 = 1.432788

d2 = 0.189269 d3 = 0.001308

The zero value is a SPEI average. Positive numbers in the study region imply above-normal precipitation, whereas negative values suggest a drought scenario (Table 3).

## Mann-Kendall trend test

The Mann-Kendall trend statistics are commonly used for analyzing trend in time series precipitation and temperature data as well as in other environmental parameters [56]. The MK trend test was performed in XLSAT Addinsoft program. The Mann-Kendall test is useful in

predicting increasing or decreasing trend in time series data.

$$S = \sum_{i=j}^{n-1} \sum_{j=i+1}^{n} sgn(xj - xi) \tag{13}$$

$$where\ sgn(xj - xi) = \begin{Bmatrix} +1, & xj > xi \\ 0, & xj = xi \\ -1, & xj < xi \end{Bmatrix} \tag{14}$$

In order to find out the trend by Tau value the following equations were used.

$$\tau = \frac{2S}{n(n-1)} \tag{15}$$

The variance of statistics denoted as.

$$Var(S) = \frac{n(n-1)(2n+5) \sum_{i=1}^{n} t_i i(i-1)(2i+5)}{18} \tag{16}$$

In the present study, the Mann-Kendall test has been conducted with a significance level of 95%. The interpretation of MK trend test is based on the p-value, where the p-value of less than 0.05 indicates the presence of a significant trend in the temporal data records. Conversely, the p-value greater than 0.05 suggests that there is no trend exist in the data.

## Results and discussion

### Spatio-statistical analysis of annual mean temperature, rainfall and Aridity Index

The average annual temperature, annual rainfall and annual average aridity index data has been analyzed to identify the minimum, maximum, mean, standard deviation (SD) and coefficient of variation (CV) during the study period. The highest minimum temperature was in Lasbella (25.64˚C) while the lowest was found in Kalat (12.32˚C). Similarly, the highest maximum temperature was found in Sibi (28.83˚C) and lowest was in Kalat (18.45˚C) while the highest standard deviation was observed at Zhob meteorological station (1.14˚C) whereas highest coefficient of variation was in Kalat (7.59%) as shown in Table 4.

Similarly, the rainfall data was checked to calculate the minimum, maximum, mean, standard deviation and coefficient of variation. Minimum rainfall was observed in Dalbandin met-station 3.50 mm during in 2000 while in the same station 7 mm rainfall was observed in 2002. In Lasbella the minimum rainfall of 8.70 mm and in Sibi 9.7 mm was the lowest rainfall in the study period during 2002. Similarly, the maximum rainfall was observed in Kalat (982.25 mm) during 1997, in Khuzdar (594.70 mm) in 1994, Sibi 499 mm in 2020 (Table 4). The years 1994, 1997, and 2020 were the wettest years in the study region. Mean highest rainfall was found in Zhob (278.04 mm) followed by Khuzdar (258.98 mm) while the lowest mean rainfall was observed at Dalbandin met-station (78.87 mm). The highest standard deviation in rainfall was observed in Kalat 163.05 mm while the highest coefficient of variation (73.35%) was also observed in the same meteorological station. Aridity index shows the climatic condition of a particular met-station i.e. is it arid, semi-arid or humid based on the derived values. Based on aridity index values, four out seven meteorological stations are arid while the remaining three semi-arid. Table 4 shows the minimum, maximum, mean, standard deviation and coefficient of variation of aridity values at each meteorological station during the study period.

**Table 4. Basic Statistics of temperature, rainfall and aridity in selected meteorological stations in Baluchistan.**

| Met-Station | Minimum Temp | Maximum Temp | Mean Temp | SD Temp | CV % Temp |
|---|---|---|---|---|---|
| Dalbandin | 21.77 | 24.97 | 23.20 | 0.75 | 3.17 |
| Kalat | 12.32 | 18.45 | 14.34 | 1.10 | 7.59 |
| Khuzdar | 19.37 | 24.83 | 22.11 | 1.06 | 4.70 |
| Lasbella | 25.64 | 28.47 | 27.16 | 0.70 | 2.53 |
| Quetta | 14.31 | 18.47 | 17.03 | 0.83 | 4.78 |
| Sibi | 24.13 | 28.83 | 27.15 | 1.02 | 3.69 |
| Zhob | 15.90 | 20.84 | 19.25 | 1.14 | 5.84 |
| Met-Station | Minimum Rainfall | Maximum Rainfall | Mean Rainfall | SD of Rainfall | CV % of Rainfall |
| Dalbandin | 3.50 | 182.04 | 78.87 | 46.27 | 57.84 |
| Kalat | 47.00 | 982.25 | 218.99 | 163.05 | 73.35 |
| Khuzdar | 89.95 | 594.70 | 258.98 | 125.81 | 47.90 |
| Lasbela | 8.70 | 474.60 | 190.18 | 110.20 | 57.09 |
| Quetta | 31.00 | 459.01 | 236.22 | 102.94 | 42.93 |
| Sibi | 9.70 | 499.06 | 205.96 | 104.20 | 49.88 |
| Zhob | 109.60 | 495.00 | 278.04 | 101.89 | 36.12 |
| Met-Station | Minimum Aridity | Maximum Aridity | Mean Aridity | SD of Aridity | CV % Aridity |
| Dalbandin | 0.39 | 0.75 | 0.47 | 0.07 | 14.71 |
| Kalat | 0.29 | 0.53 | 0.34 | 0.04 | 12.71 |
| Khuzdar | 0.29 | 0.71 | 0.39 | 0.06 | 16.42 |
| Lasbela | 0.24 | 0.32 | 0.28 | 0.02 | 6.97 |
| Quetta | 0.34 | 0.61 | 0.41 | 0.05 | 11.36 |
| Sibi | 0.35 | 0.53 | 0.44 | 0.04 | 8.22 |
| Zhob | 0.36 | 0.53 | 0.43 | 0.03 | 7.75 |

## Spatial distribution of annual rainfall, Potential Evapotranspiration (PET) and Aridity Index ($A_I$)

It is observed that in all of eight met-stations the values of annual rainfall, PET and aridity index vary with topography. The monsoon-region stations received more rainfall than those that rely primarily on the winter disturbances. The stations that received the highest amount of annual rainfall are Zhob (278.2mm), Khuzdar (258.9mm), Quetta (236.2mm), and Kalat (218.1mm). The lowest mount of rainfall receiving met-stations are Dalbandin (81.4mm), Sibi (207.3mm) respectively (Table 5).

The analysis shows that PET values are higher in low-elevated areas where the temperature is high than the high-elevated regions having low temperature. The maximum PET values was

**Table 5. Climatic characteristics in selected meteorological stations in Baluchistan province.**

| Met-stations | Annual Rainfall (mm) | Annual PET (mm) | Aridity Index | SD of Annual PET | Climate (UNEP 1997) |
|---|---|---|---|---|---|
| Dalbandin | 81.4 | 1708.8 | 0.0476 | 211 | Arid |
| Kalat | 218.1 | 769.7 | 0.2834 | 33.7 | Semi-arid |
| Khuzdar | 258.9 | 1463.8 | 0.1769 | 626 | Arid |
| Lasbella | 190.1 | 2353.8 | 0.0808 | 261.7 | Arid |
| Quetta | 236.2 | 934 | 0.2529 | 51.8 | Semi-arid |
| Sibi | 207.3 | 3063.6 | 0.0677 | 555.4 | Arid |
| Zhob | 278.2 | 1111.8 | 0.2502 | 100.2 | Semi-Arid |

recorded in Sibi (3063.6 mm), Lasbella (2353.8 mm), Dalbandin (1708.8 mm), and Khuzdar (1463.8 mm), on the other hand Kalat (769.7 mm) and Quetta (934 mm) have low PET due to low temperature and high elevations. The standard deviations were calculated for annual mean PET. The maximum value of standard deviation noted in Khuzdar followed by Sibi and Lasbella.

The aridity index is an important indicator of climate severity, as it reflects the balance between precipitation and evapotranspiration. Aridity Index is the degree of dryness at a particular area [57]. The $A_I$ is derived using the annual precipitation to potential evapotranspiration [58]. According to Hussain and Hussain [49], the $A_I$ value ranges from 0.03 to 0.2 indicate arid climates, 0.21 to 0.5 as semi-arid, 0.51 to 0.65 sub-humid, and the $A_I$ value above 0.65 indicates humid regions. The $A_I$ results revealed that arid to semi-arid conditions prevails in all meteorological stations of the Baluchistan province. The meteorological stations present in mountainous regions exhibits semi-arid condition i.e. in Kalat, Quetta, and Zhob whereas, Sibi, Lasbella and Khuzdar stations are experiencing arid climate in Baluchistan province (Fig 2 and Table 5).

The linear regression was applied to determine variation in $A_I$. The Fig 3 shows that the aridity index shows variation from area to area as some of the meteorological stations observed

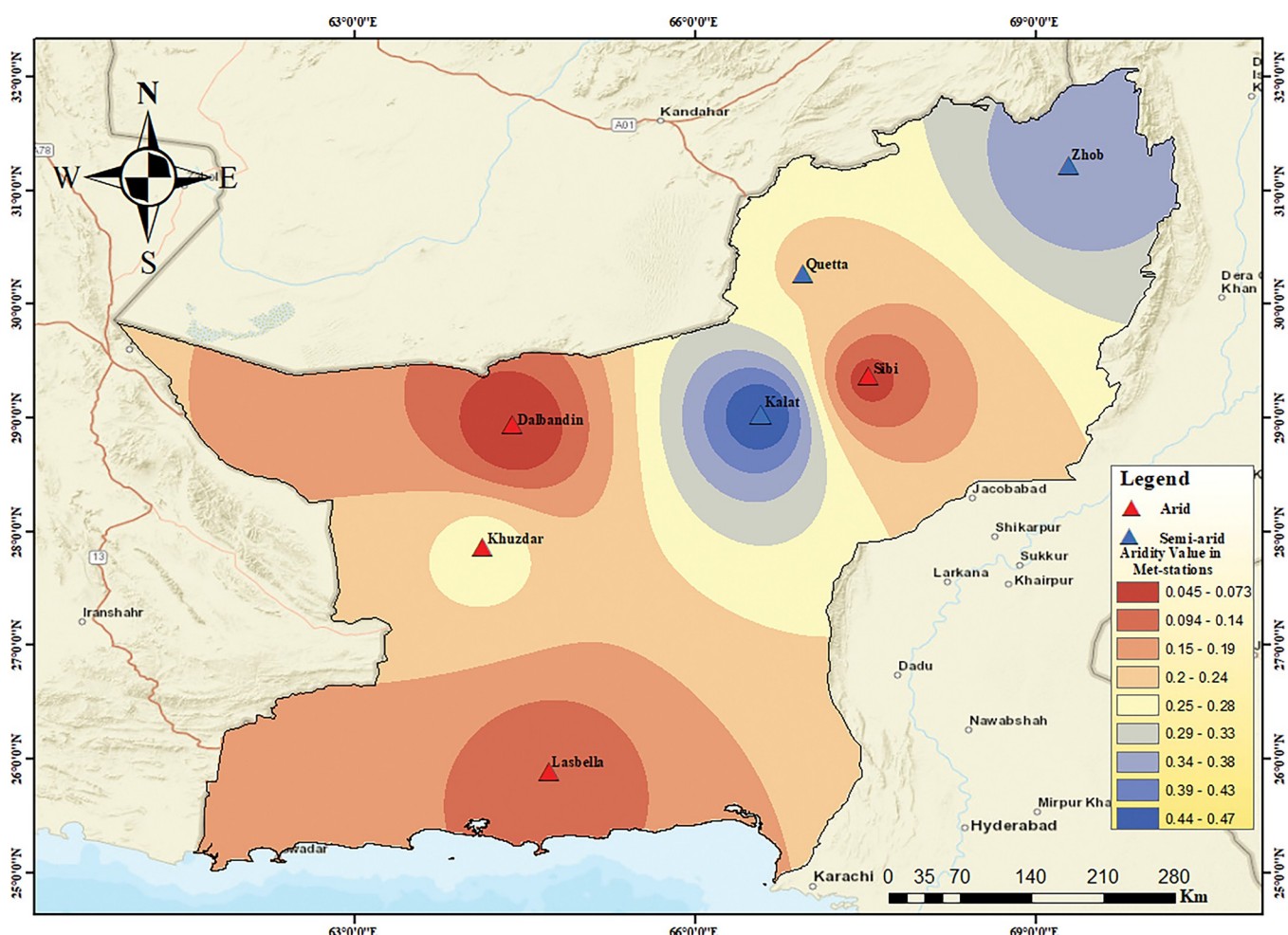

**Fig 2. Spatial Distribution of Aridity Index in the selected Met-stations of Baluchistan.**

increasing while in some the aridity index shows decreasing behavior in linear regression results. The met-stations in which the aridity is increasing are Kalat (0.0065/year), Quetta (0.0025/year), Zhob (0.002/year) Khuzdar (0.0018/year), and Dalbandin (0.0005/year) while the decreasing aridity is noted in Sibi (0.0015/year), and Lasbella (0.0002/year). In majority of the met-stations of Baluchistan the $A_I$ observed increasing trend (Fig 3). Increasing aridity will increase the threat of drought as in the study area the climate is already arid to semi-arid. Low precipitation and high evapotranspiration are the main factors of increasing aridity index

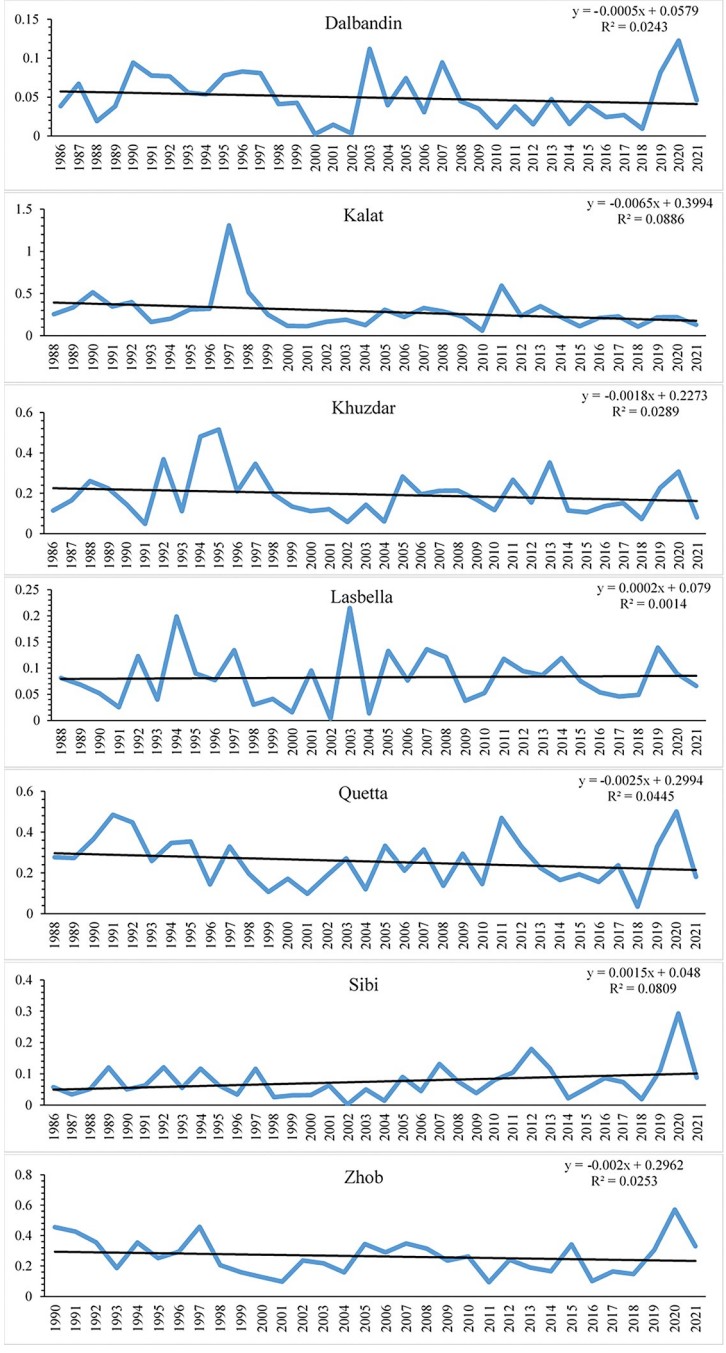

**Fig 3. The temporal variation of Aridity index in Baluchistan province.**

values and it will lead to more extreme drought conditions. It is observed in all met-stations the most affected period is from 1998–2004 and followed by 2010 (Fig 3).

## Spatio-temporal variation of 1 and 6-Months SPEI

The 1 and 6-month SPEI was calculated for the selected met-stations to find out the drought conditions in the study area. Both 1 and 6-month SPEI shows extreme, severe and moderate drought events in the study area (Fig 4). Six extreme drought events were observed 1 and 6-month SPEI in Dalbandin, Khuzdar, Lasbella and Zhob. In Dalbandin (1-month SPEI, 1994,2006,2011,2020, and 6-Month SPEI, 2019,2021), Khuzdar (1-month SPEI,1999,2003,2012,2019, and 6-Month SPEI 1991,2015), Lasbella (1-month SPEI 1997,1998,2001,2005,2018,2021, and 6-Month SPEI 2018) and Zhob (1-month SPEI, 2004,2013,2016,2017,2018, and 6-Month SPEI, 2000). The extreme drought in Kalat (1-month SPEI, 1993,2005,2012, 6-month SPEI, 2019,2021, and 6-Month SPEI,) and Quetta (1-month SPEI, 1994,2006,2019, and 6-month SPEI, 2018) and Sibi (1-month SPEI, 2004,2010,2015,2019, and 6-Month SPEI 2002,2020). In 1- month SPEI the extreme droughts were more than 6-month SPEI in all met-stations. The stations that are the most affected by severe drought conditions in 1 and 6-month are the Quetta (1-month SPEI 1997,1998,2001, 2006,2008,2010,2018, and 6-Month SPEI, 2002,2014,2015,2016), Sibi (1-month SPEI 1987,2003,2014,2015,2021, and 6-Month SPEI, 1988,1998,2000,2021) Lasbella (1-month SPEI 1990, 2006,2009,2010,2013, and 6-Month SPEI 1999,2002,2019,) Dalbandin (1-month SPEI, 2011,2014, and 6- Month SPEI 2000,2010,2017), Zhob (1-month SPEI, 1997,2006,2009, 2019,2021 and 6-Month SPEI, 2010) and Khuzdar (1-month SPEI,1990,2006,2018 and 6-Month SPEI, 1986). While moderate drought was found in most of the years in all met-station for 1 and 6-month SPEI (Fig 4). Quetta, Sibi, and Zhob met-station were affected from moderate to severe drought in 1998–2003. The frequency of drought by its percentage is calculated for 1 and 6-month SPEI of all met-stations. The drought frequency in Lasbella 46% of extreme-severe drought followed by Zhob (37%), Quetta and Sibi (34%), Dalbandin (31%), Kalat and Khuzdar (28%) as shown in Table 6. The result of 1 and 6-month SPEI clearly indicated that the most common spells of extreme to severe drought conditions in most of the stations are observed in 1998–2003, 2006, 2010, 2015–2016 and 2019 (Table 6). While the common extreme wet-moderate conditions were observed from 1992, 1993, 1994, 1995, 2011 and 2012 for most of the stations (Fig 4).

The worse spell of drought observed during 1998–2002 and 2014–2018 affected livestock, agriculture, and crops [27,30,40]. Extreme drought was experienced in Baluchistan province during 1998 to 2003. This prolonged drought depleted groundwater table, resulted in agricultural crop failure and famine across the province. It was Pakistan's worst drought of Pakistan, affecting over 3.3 million people. According to the SPEI 1 and 6-month results, Quetta, Dalbandin, and Kalat, observed successive dry events following the 1998–2003 drought. The consective droughts are observed in winter rainfall receiving met-stations (Quetta, Dalbandin and Kalat) as compared to monsoon rainfall receiving met-stations. Mostly drought was observed was. The drought frequency and intensity increased during the study period, which was connected to decline in precipitation and increase in temperature in the study area.

## Drought characteristics

In the Baluchistan province, the longest drought duration was observed in Kalat followed by Dalbandin and Khuzdar meteorological stations in 2021 with drought severity -20.96 in Kalat, -14.11 in Dalbandin and -11.92 in Khuzdar (Figs 5 and 6). Intensity of drought was also found

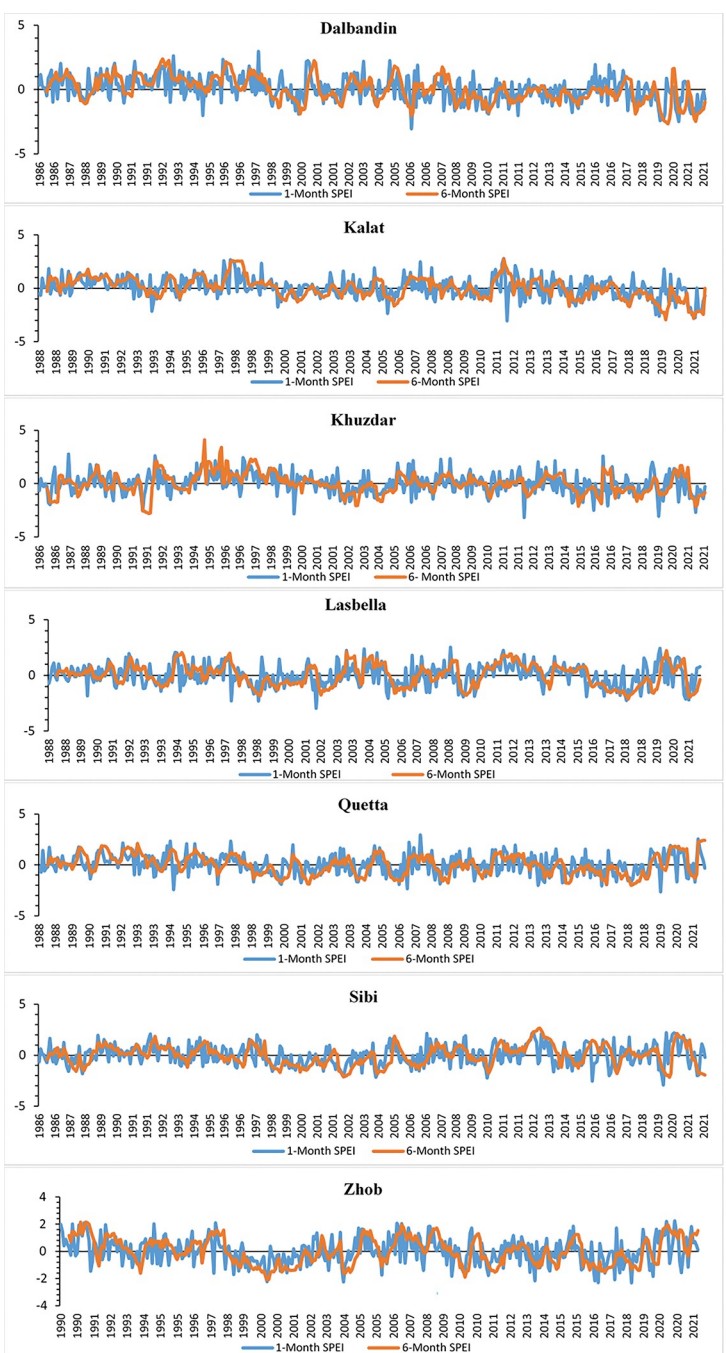

**Fig 4. The temporal distribution of 1 and 6-Month SPEI in selected meteorological stations of Baluchistan province.**

high in 2021 in these three meteorological stations which were -2.095, -2.015 and -1.986 in Kalat, Dalbandin and Khuzdar respectively (Fig 7). The meteorological stations Quetta and Lasbella observed highest duration of drought in 2018 (Fig 5) with drought severity of -965 and -12.46 respectively (Fig 6) whereas drought intensity in Quetta was -1.609 and in Lasbella was -2.077 (Fig 7).

**Table 6. The meteorological stations of Baluchistan with extreme to severe drought.**

| Met-stations | Extreme Drought Spell 1 and 6-Month | Severe Drought Spell 1 and 6-Month | Percentage of Drought Spell 1 and 6-Month |
|---|---|---|---|
| Dalbandin | 6(1994,2006,2011,2019,2020, 2021) | 5(2000,2010,2011,2014,2017) | 31% |
| Kalat | 5(1993,2005,2012,2019,2021) | 4(2000,2012,2015,2018) | 28% |
| Khuzdar | 6(1991,1999,2003,2012,2015,2019) | 4(1986,1990,2006,2018) | 28% |
| Lasbella | 6(1997,1998,2001,2005,2018,2021) | 9(1990,1999,2002,2006,2009,2010,2013,2016,2019) | 46% |
| Quetta | 4(1994,2006,2018,2019) | 8(1997,1998,2000,2002,2010,2015,2016,2018) | 34% |
| Sibi | 5(2002,2004,2010,2015,2019) | 7(1987,1988,1998,2000,2014,2015, 2019) | 34% |
| Zhob | 6(2000,2004,2013,2016,2017,2018) | 7(1994,1997,2006,2009,2010,2019,2021) | 37% |

## Monthly $A_I$ and 1-month SPEI

The SPEI and the Aridity Index have strong relationship, and the $A_I$ can be used to forecast the SPEI in a specific region. The MK trend test was applied to monthly $A_I$ and 1-month SPEI to find out the significant trends in both time series results. The p-value of less than 0.05 shows the existence of a significant trend in the temporal datasets, while a p-value greater than 0.05 indicates the absence of trend in the data. For further details on the severity of $A_I$ and drought, the Kendall tau value was showed on maps using inverse distance weight (IDW) technique (Figs 8 and 9). Positive numbers indicates increasing trend, whereas negative values reflect decreasing trend over a specific time period and area. Seasonal analysis was performed and the monthly data was accumulated in seasons like winter (Dec, Jan, Feb), spring (March, April, May), summer (Jun, July, Aug), and autumn (Sep, Oct, Nov). The MK trend test results of the monthly aridity index and 1-month SPEI indicated variation across the study area. The results of $A_I$ showed both positive and negative trends, while the results of 1-month SPEI showed negative trend in all seasons. The positive trend in $A_I$ is noted in winter (Dec, Jan, Feb) in Dalbandin and Lasbella stations which indicates increasing dry conditions. While in Zhob stations the negative tau value shows decreasing dry condition in month of January. Spring, particularly March and April, shows a significant decreasing trend in Khuzdar and Lasbella, which indicates comparatively wet conditions are prevailing. On the other hand, Kalat showed a significant positive trend in March, which means an increase in $A_I$. Whereas in summer, Kalat meteorological station shows a positive trend showing an increasing dry condition in the station. The most increasing wet conditions were found in October in Dalbandin, Khuzdar, Lasbella, and Quetta (Fig 8).

Similarly, the MK trend test results for 1-month SPEI were calculated, and all meteorological stations showed significant negative trend in some months. The results showed that that in all seasons, the Kendall tau-value was found negative and the p-value was found less than 0.05, indicating increasing trend of drought in the study area. The significant negative (tau-value) trend is noted in spring, where in March and May the drought is decreasing in Dalbandin, Kalat, and Khuzdar. Besides Dalbandin, Kalat, Khuzdar, and Lasbella showed a significant negative trend in the month of April. This indicates an increase in precipitation in the spring season in these specific meteorological stations. The Dalbandin, Kalat, and Quetta met-stations showed significant negative trend in the months of December and February of the winter season. In January, the Dalbandin, Kalat, Quetta, and Sibi met-stations exhibited a significant negative trend in the 1-month SPEI which indicated decrease in drought and increase in rainfall. In summer, the Kalat and Quetta stations witnessed a significant negative trend in the months of June and July, while August showed no trend in the 1-month SPEI. In autumn, all three months of September, October, and November exhibited significant negative trends at a few meteorological stations, such as Dalbandin, Kalat, and Khuzdar (Fig 9). In the results of

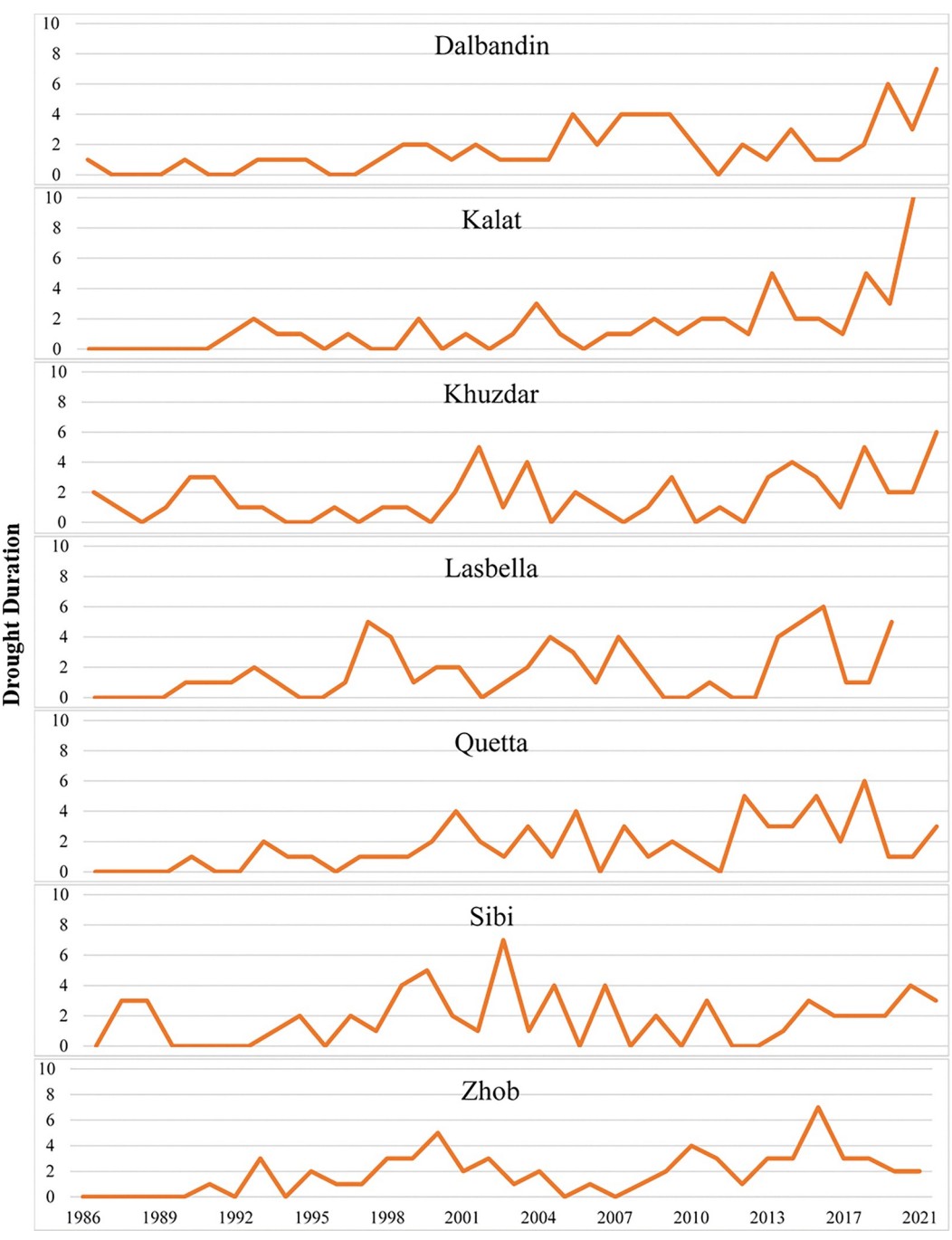

**Fig 5. Drought duration in selected meteorological stations of Baluchistan province.**

MK for 1-month SPEI in every season, a negative trend is noted that indicates the increase in rainfall and decrease in drought spells in all those years (1986–2021).

## Discussion

The meteorological drought assessment was carried out in this study using *AI* and SPEI indices that identified a number of drought spells during the study period in Baluchistan province.

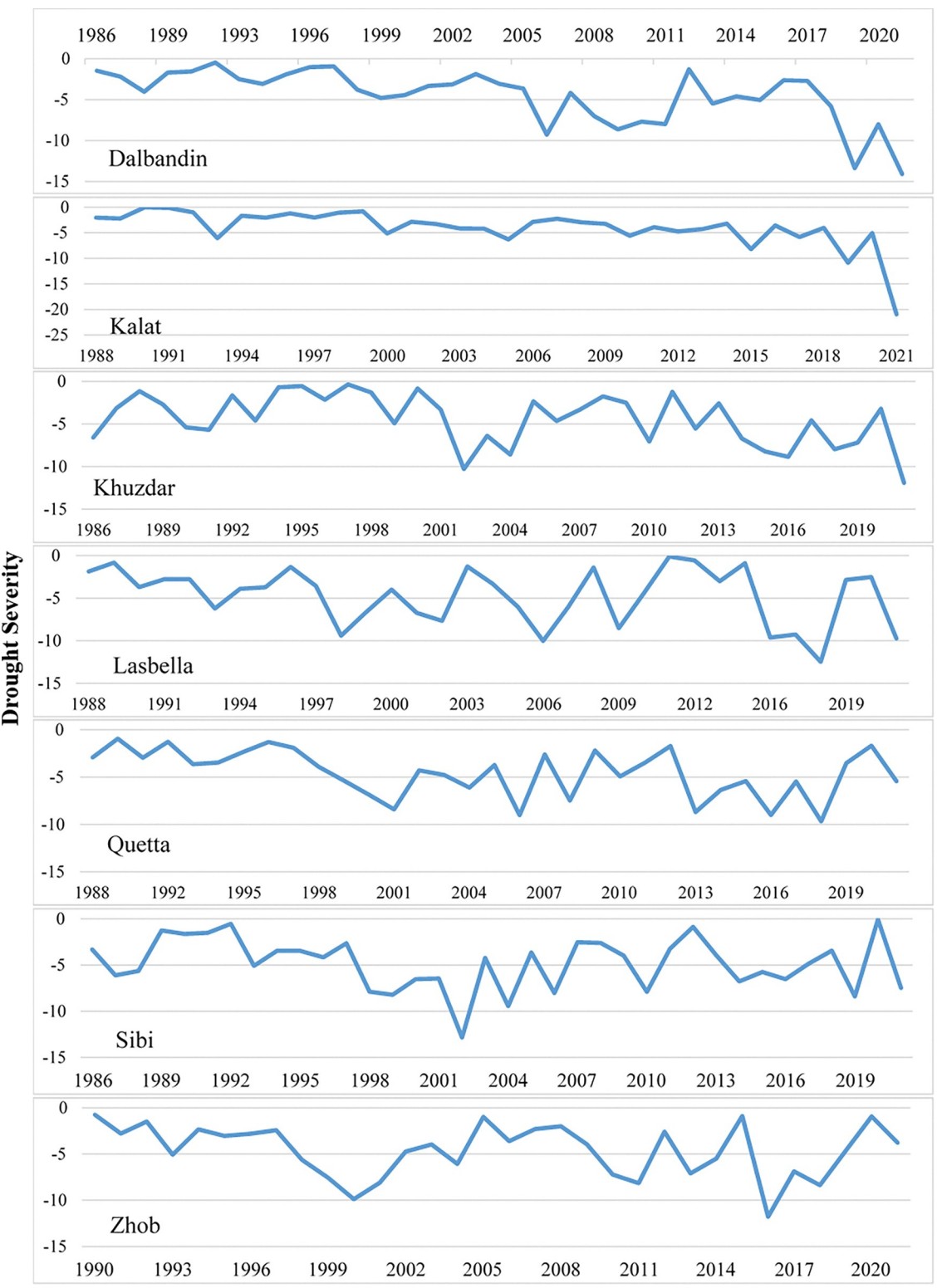

**Fig 6. Drought Severity in selected meteorological stations of Baluchistan province.**

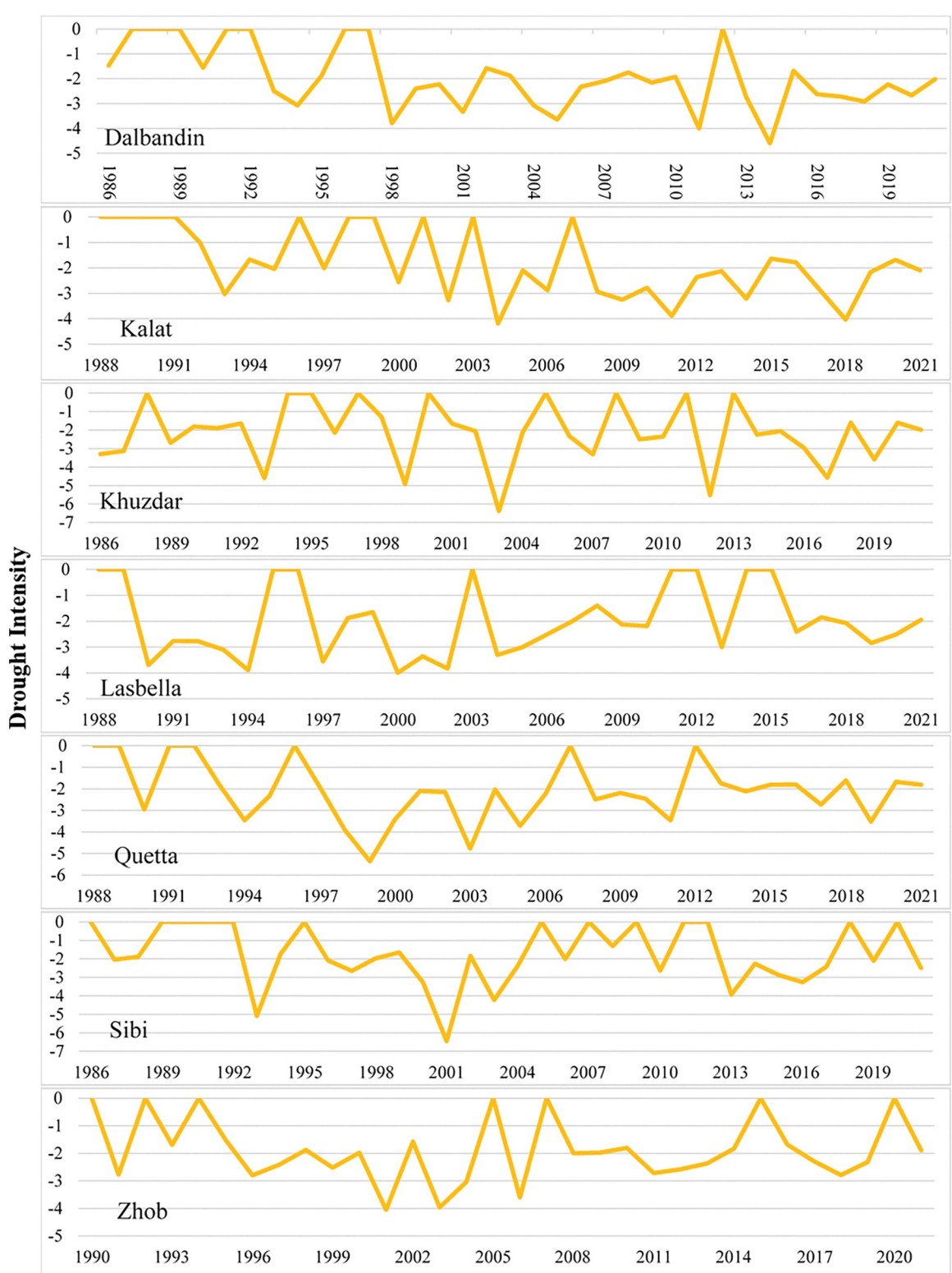

**Fig 7. Drought Intensity in selected meteorological stations of Baluchistan province.**

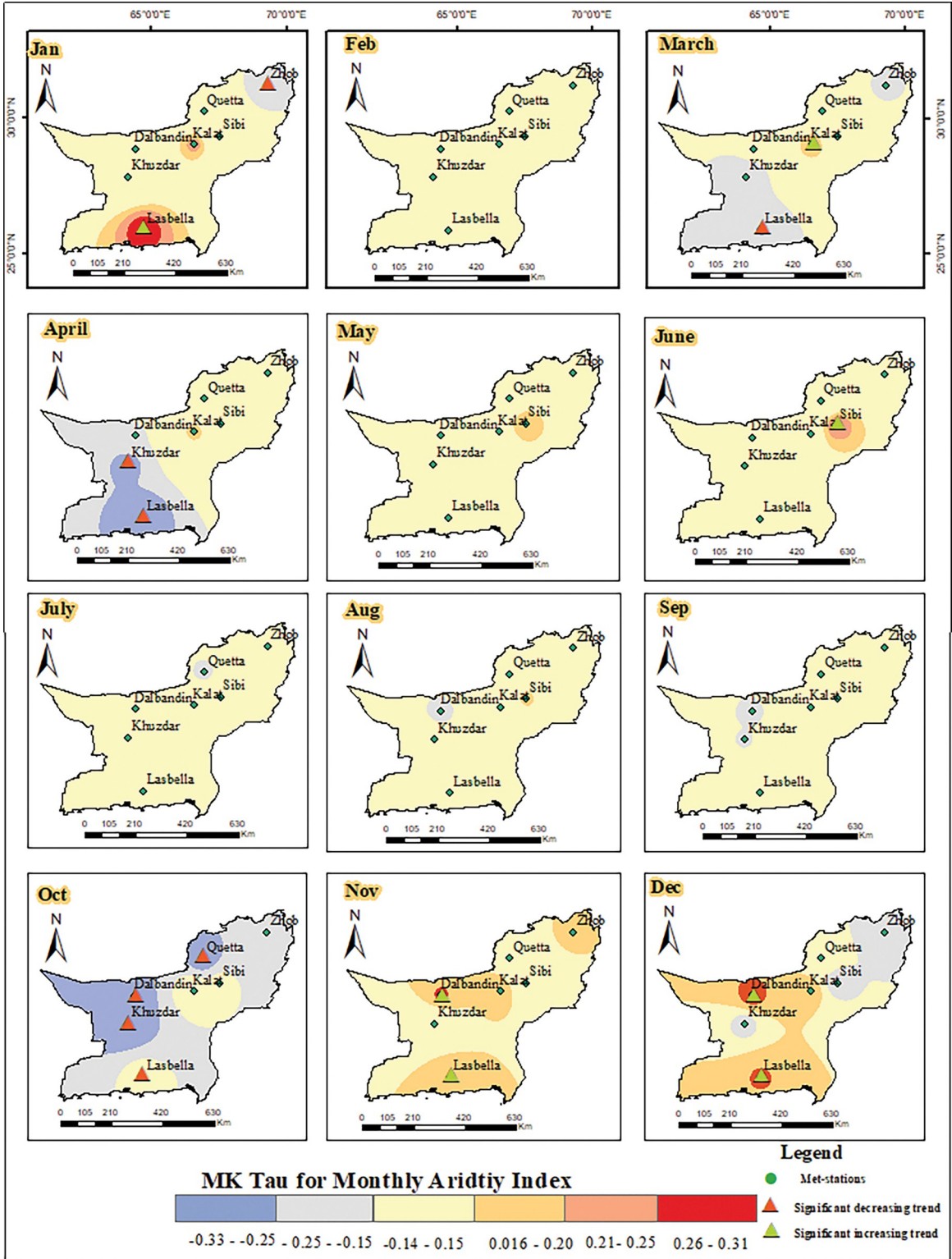

**Fig 8. *Distribution of MK trend values for* monthly Aridity Index in Baluchistan Province.**

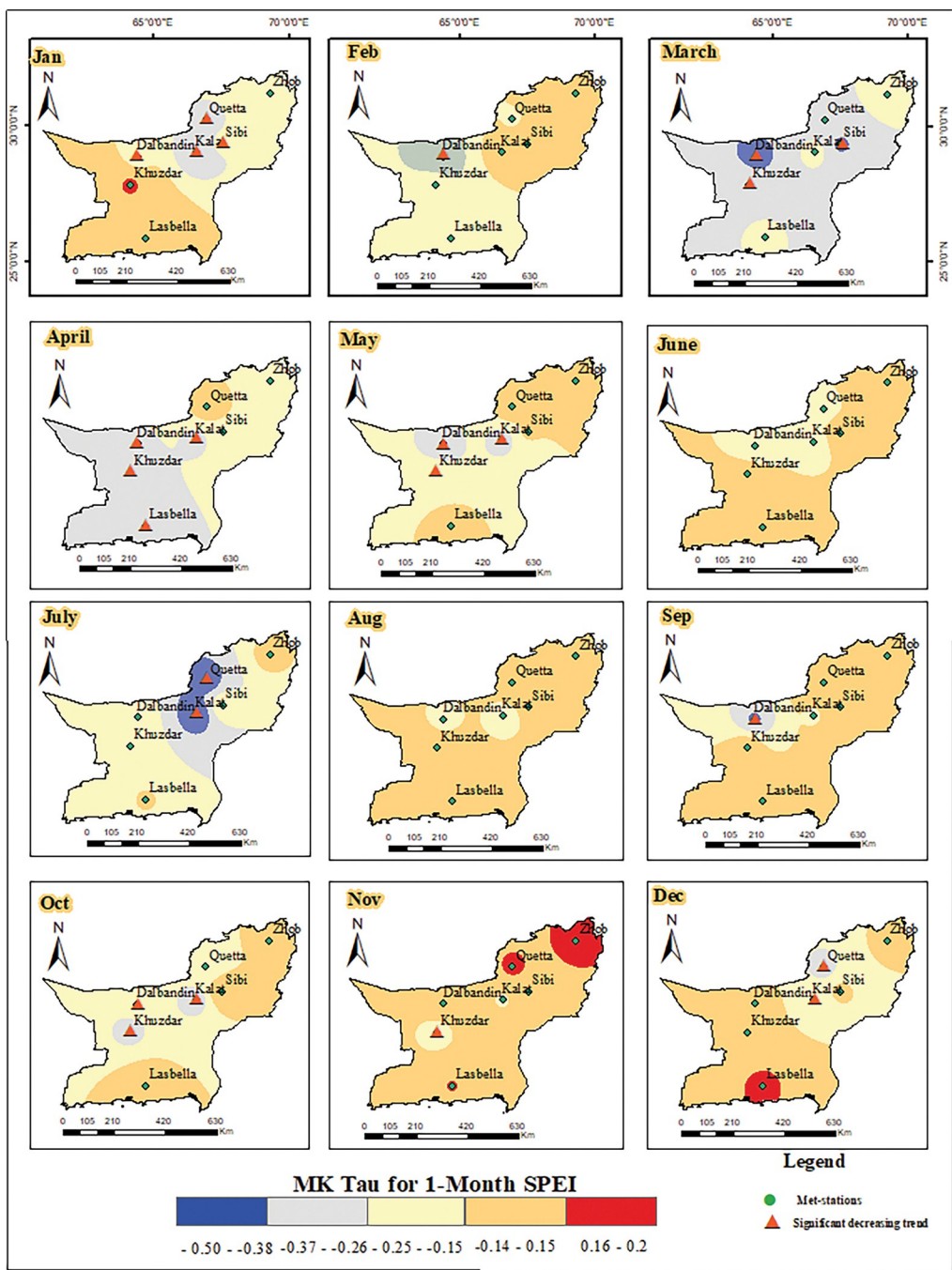

**Fig 9.** *Distribution of MK trend values for* 1-Month SPEI in Baluchistan province.

The major cause of these cyclic drought events in the study region is El-Nino Southern Oscillation (ENSO) responsible for low precipitation and make the region vulnerable to drought hazards [59,60]. Baluchistan province is already prone to droughts due to its arid and semi-arid climate and the increasing aridity and drought trend makes it more vulnerable to such hazards. There are also some local causes of decreasing rainfall and increasing droughts in the region i.e. deforestation for fuel wood, agricultural expansion on natural vegetation covered areas, mining, construction and built-up area expansion especially in the mountainous areas that

cause increase in temperature and decrease in precipitation and ultimately leads to aridity as well as drought. Aridity Index identify wither the meteorological station is arid, semi-arid or humid and the temporal study of this index identify either this aridity level is increasing or decreasing with the passage of time while SPEI specifically identify drought characteristics in the study region using both rainfall and temperature influences. In this study, we only consider the 1 and 6-month SPEI as both are suitable for short time drought assessment. This study observed prolonged extreme drought in 1998–2002 and in the same period prolonged and severe droughts were prevailed all over Asia due to oceanic and atmospheric variability over Pacific and Indian ocean which affect the monsoonal region and caused prolonged droughts [61–63]. The results of the study identified increasing trend in Aridity Index especially in winter season which means that rainfall in the study region are decreasing in winter and the same decreasing rainfall pattern in winter as well as spring season was observed by Shelton and Dixon [64] in Baluchistan and Sindh province. Aamir et al. [65], also observed decreasing trend in winter and spring precipitation in Baluchistan province during 1977 to 2017 and which is the main reason of increasing aridity and drought in this region. Similarly, the study results shows that during 1999–2002 and 2014–2018 were continuous dry period in Baluchistan province and its surroundings and the same period was observed in another study conducted by Shelton and Dixon [64]. The SPEI results of 1 and 6-months identified drought spells during 1998–2004, 2006, and 2014–2018 in other region of Pakistan and in Asia [5,13,14,27,66–69]. On the basis of drought frequency of 1 and 6-month SPEI, the maximum drought frequency was observed in Lasbella (46%), followed by Zhob (37%), Quetta and Sibi both (34%), and Dalbandin (31%). In Kalat and Khuzdar, the drought frequencies are lower than in the other stations (28%). These high frequency of drought not only affect the people of the province but also agriculture and livestock sectors as well as surface and groundwater table [30]. As the study area already experiencing arid and semi-arid climate therefore a little variation in rainfall can cause extreme dry conditions in the province and thus drought effects are observed on various sector of life [70]. In Pakistan, monsoon rainfall varies significantly in terms of timing of onset, location, duration, and intensity of rainfall, thus the delay in onset timing and duration of monsoon also became a cause of drought in Baluchistan province [71,72].

Similar drought trend and drought pattern has been observed in recent study conducted by Naz et al. [25]. The increasing global temperature due to climate change is the main driving force of changing rainfall pattern over Pakistan [73]. The variability in rainfall pattern and trend is one of the major challenge in Baluchistan province as the region is already facing water scarcity issues due to its arid and semi-arid climate. The variation in rainfall trend caused frequent droughts spells of various duration in last three decades and the drought spell during 1998–2002 was the history worst drought which affected millions of people and brought famine in the area [25,29,73]. To combat the issue of drought in the region, climate-resistant crops must be introduced, drip irrigation system should be introduced in the area and the capability and system of karez irrigation should be enhanced to use water resources in sustainable way. The findings of this study lead to the conclusion that Baluchistan will likely to experience the repeated drought events in future. Such dry spells might have serious impacts on agricultural, hydrological, and social systems. In combating climate change issues, the government has recently launched the ten billion trees tsunami project in Pakistan which may have favorable impact on the local climate. To examine mitigation and adaptation options, it is vital to do micro-level research on the evaluation of drought consequences. It is important for policymakers and stakeholders to take immediate measures to address this issue and mitigate its effects, which may include implementing sustainable water management practices, promoting drought-resistant crops, and investing in alternative livelihoods.

## Conclusion

This focused on the variability of aridity and drought in the arid to semi-arid region of Pakistan (Baluchistan). The UNEP 1997 climate aridity index was used to calculate the aridity for each met-station. Out of eight meteorological stations, the aridity is increasing at five stations. The stations with increasing aridity are Kalat, Quetta, Zhob, Khuzdar, and Dalbandin. The highest rate of increasing aridity observed at Kalat (0.0065/annum). The results of 1-month and 6-month SPEI identified the periods of extreme to severe drought in most of the stations were in 1998–2003, 2006, 2010, 2015–2016, and 2019 in the study area. In the study area, the summer monsoon areas are more affected as compared to the winter. The drought frequency at Lasbella was 46% in 1-month and 6-month SPEI. The frequency of drought events was 28 percent in Kalat and Khuzdar meteorological stations. The drought characteristics showed longest drought duration, highest intensity and more svereity in Kalat meteorological station followed by Dalbandin and Khuzdar. The MK trend test results showed that the monthly $A_I$ is increasing in the month November to January at Dalbaindin and Lasbella while in June the Kalat shows increasing aridity. The monthly $AI$ and 1-month SPEI both showed decreasing trend in majority of the stations in the months of April and October.

## Acknowledgments

We the author are acknowledge the support of Pakistan Meteorological Department for providing climatic data for this research study. We are thankful to Dr. Jan Bloemendal for English language correction.

## Author Contributions

**Data curation:** Hammed Ullah Khan.

**Formal analysis:** Yanpei Cheng, Yuanjie Zhao.

**Funding acquisition:** Yue Cong Li.

**Methodology:** Ghani Rahman.

**Software:** Hammed Ullah Khan.

**Supervision:** Yue Cong Li, Ghani Rahman.

**Writing – original draft:** Muhammad Rafiq.

**Writing – review & editing:** Ghani Rahman.

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
