## [Decision Letter · Decision Letter 0]

18 Jun 2023

PONE-D-23-15469Estimation of Regional Meteorological Aridity and Drought Characteristics of Baluchistan Province, Pakistan ”PLOS ONE

Dear Dr. Muhammad Rafiq,

Thank you for submitting your manuscript to PLOS ONE. After careful consideration, we feel that it has merit but does not fully meet PLOS ONE’s publication criteria as it currently stands. Therefore, we invite you to submit a revised version of the manuscript that addresses the points raised during the review process.

We look forward to receiving your revised manuscript.

Kind regards,

Salim Heddam

Academic Editor

PLOS ONE

Journal Requirements:

"This study is supported by the National Natural Science Foundation of China (Grant Nos. 41877433) and the Hebei Natural Science Foundation and Key Basic Research (18963301D)."

"Specify the role(s) played"

"NO authors have competing interests"

7. PLOS requires an ORCID iD for the corresponding author in Editorial Manager on papers submitted after December 6th, 2016. Please ensure that you have an ORCID iD and that it is validated in Editorial Manager. To do this, go to ‘Update my Information’ (in the upper left-hand corner of the main menu), and click on the Fetch/Validate link next to the ORCID field. This will take you to the ORCID site and allow you to create a new iD or authenticate a pre-existing iD in Editorial Manager. Please see the following video for instructions on linking an ORCID iD to your Editorial Manager account: https://www.youtube.com/watch?v=_xcclfuvtxQ

8. Please amend your list of authors on the manuscript to ensure that each author is linked to an affiliation. Authors’ affiliations should reflect the institution where the work was done (if authors moved subsequently, you can also list the new affiliation stating “current affiliation:….” as necessary).

9. Please include your full ethics statement in the ‘Methods’ section of your manuscript file. In your statement, please include the full name of the IRB or ethics committee who approved or waived your study, as well as whether or not you obtained informed written or verbal consent. If consent was waived for your study, please include this information in your statement as well. 

10. We note that Figures 1,2, 5 and 6 in your submission contain [map/satellite] images which may be copyrighted. All PLOS content is published under the Creative Commons Attribution License (CC BY 4.0), which means that the manuscript, images, and Supporting Information files will be freely available online, and any third party is permitted to access, download, copy, distribute, and use these materials in any way, even commercially, with proper attribution. For these reasons, we cannot publish previously copyrighted maps or satellite images created using proprietary data, such as Google software (Google Maps, Street View, and Earth). For more information, see our copyright guidelines: http://journals.plos.org/plosone/s/licenses-and-copyright.

a. You may seek permission from the original copyright holder of Figures 1,2, 5 and 6 to publish the content specifically under the CC BY 4.0 license.  

Additional Editor Comments:

Reviewer 1:

The paper is well organized and well written according to the journal format. English language is good however, I have following concerns before accepting this manuscript.

Dear authors, I found some grammatical and technical issues in the abstract section therefore I will suggest to rewrite and align it with international journal standards.

The given keywords are only three in numbers and there should be at least five keywords.

I will suggest to improve the writing of the introduction section and also include some further latest and updated literature regarding drought and aridity in the region.

In the study area section “From a morphological perspective, the study area (Baluchistan province) is a mountainous, arid in southwest territory of Pakistan (Fig .1).” morphology is not an accurate word. Replace it with some more suitable word.

Correct the given latitude and longitude in the study area section “The region is located between 30.12 oN and 67.01 oE latitude and longitude, respectively.”

There is a lot of grammatical and technical mistakes in the study area section like (i) “and are covered in dry terrain, are west of the Baluchistan plains.” (ii) “The bulk of the province is covered by the Kharan desert in the southwest” (iii) “Additionally, the province is crucial geographically for both regional trade and security.” (iv) “Baluchistan makes up 4% of Pakistan's overall land mass” (v) “Due to terrain brought on by monsoon and western storms, there are variations in rainfall and temperature dispersion.” (vi) “Baluchistan primarily experiences erratic weather that ranges from 30 mm to 397 mm annually”.

Figure 1 needs to be corrected as it does not reflect the actual topography of the study area.

Rewrite the first sentence of the data collection and method section.

Rewrite the caption of table 1.

Rewrite the caption of table 2

If this manuscript is based on SPEI then why the authors are focusing on SPI in the methodology section.

Rewrite table 3 caption

All equations must be numbered in sequential order.

Separate the Results and then write a separate discussion instead of results and discussion.

Rewrite the caption of table 4

Rewrite the caption of table 5

Rewrite the caption of Figure 4.

Rewrite the caption of Figure 5.

The discussion section needs to be carefully studied and revised in light of more scientific literature with analytical reasoning.

The conclusion section needs proofreading for grammatical correction.

Reviewer 2:

The manuscript titled “Estimation of Regional Meteorological Aridity and Drought Characteristics of Baluchistan Province, Pakistan” is evaluated and the comments are mentioned below:

The work in the manuscript titled “Estimation of Regional Meteorological Aridity and Drought Characteristics of Baluchistan Province, Pakistan” used the already well-established methods for estimating characteristics of drought and aridity. The introduction section of the manuscript is not written well and there are a lot of confusing statements which make it very difficult to understand. Some of the issues are mentioned under the Specific Comments at the end of this document. The overall text of the manuscript is very confusing and not clear for understanding. I think the authors wrote the manuscript in a rush. The Discussion section of the manuscript discussed the results and therefore it is not look like a scientific discussion. I will suggest the authors to take their time and improve the overall manuscript.

Specific Comments:

1. Page 1, L 24-25: This statement “The temperature and precipitation data were obtained from the Pakistan Meteorological Department” make no sense abstract.

2. Page 1, L 29-30: This statement “The linear regression was used to find out the increasing aridity is increasing in the region” is confusing. Rephrase this for easy understanding.

3. Page 1, L 30-31: The statement “The result revealed that most stations are arid to semi-arid, and the highest increasing aridity is noted in Kalat (0.0065/annum)” can be like “The result revealed that most of the stations are arid to semi-arid, and the highest increasing aridity is noted in Kalat (0.0065/annum)”. This is one of the example of grammatical issues. Please check your manuscript thoroughly for such type of issues. I suggest to edit your manuscript from native English speaker before publication.

4. Keywords: A keyword “Aridity index AI” is mentioned in the list of keywords. It is not clear. Please mentioned clear keyword instead of confused keywords.

5. Page 2, L 56-58: The statements “The severe effects of drought are the negative effects on society and agriculture [8]. According Ofipcc (2013) defined drought as the time of unusually dry weather long enough to create a severe hydrological imbalance” are not clear. In the first statement you are talking about negative effects. What are these negative effects. The second statement is confusing in the start and not understandable.

6. Page 3, L 71-72: The statement “The Asian counties …”. I think you mean countries. Check your manuscript thoroughly for such type of issues.

7. Page 3, L 76-78: The statement “Less than 250 mm of rain falls annually throughout the entire province of Sindh, the majority of Balochistan, large portions of the Punjab, and the central regions of the Northern Areas [16]” is not clear. Rephrase it and make it clearer for easy understanding.

8. Page 8, L 178: Eq. (6) is not clear. Please check all equations of your manuscript thoroughly.

Reviewers' comments:

Reviewer's Responses to Questions

**Comments to the Author**

1. Is the manuscript technically sound, and do the data support the conclusions?

Reviewer #1: Yes

Reviewer #2: Partly

2. Has the statistical analysis been performed appropriately and rigorously? 

Reviewer #1: Yes

Reviewer #2: Yes

3. Have the authors made all data underlying the findings in their manuscript fully available?

Reviewer #1: Yes

Reviewer #2: Yes

4. Is the manuscript presented in an intelligible fashion and written in standard English?

Reviewer #1: Yes

Reviewer #2: No

5. Review Comments to the Author

Reviewer #1: The paper is well organized and well written according to the journal format. English language is good however, I have following concerns before accepting this manuscript.

Dear authors, I found some grammatical and technical issues in the abstract section therefore I will suggest to rewrite and align it with international journal standards.

The given keywords are only three in numbers and there should be at least five keywords.

I will suggest to improve the writing of the introduction section and also include some further latest and updated literature regarding drought and aridity in the region.

In the study area section “From a morphological perspective, the study area (Baluchistan province) is a mountainous, arid in southwest territory of Pakistan (Fig .1).” morphology is not an accurate word. Replace it with some more suitable word.

Correct the given latitude and longitude in the study area section “The region is located between 30.12 oN and 67.01 oE latitude and longitude, respectively.”

There is a lot of grammatical and technical mistakes in the study area section like (i) “and are covered in dry terrain, are west of the Baluchistan plains.” (ii) “The bulk of the province is covered by the Kharan desert in the southwest” (iii) “Additionally, the province is crucial geographically for both regional trade and security.” (iv) “Baluchistan makes up 4% of Pakistan's overall land mass” (v) “Due to terrain brought on by monsoon and western storms, there are variations in rainfall and temperature dispersion.” (vi) “Baluchistan primarily experiences erratic weather that ranges from 30 mm to 397 mm annually”.

Figure 1 needs to be corrected as it does not reflect the actual topography of the study area.

Rewrite the first sentence of the data collection and method section.

Rewrite the caption of table 1.

Rewrite the caption of table 2

If this manuscript is based on SPEI then why the authors are focusing on SPI in the methodology section.

Rewrite table 3 caption

All equations must be numbered in sequential order.

Separate the Results and then write a separate discussion instead of results and discussion.

Rewrite the caption of table 4

Rewrite the caption of table 5

Rewrite the caption of Figure 4.

Rewrite the caption of Figure 5.

The discussion section needs to be carefully studied and revised in light of more scientific literature with analytical reasoning.

The conclusion section needs proofreading for grammatical correction.

Reviewer #2: Review Report

The manuscript titled “Estimation of Regional Meteorological Aridity and Drought Characteristics of Baluchistan Province, Pakistan” is evaluated and the comments are mentioned below:

The work in the manuscript titled “Estimation of Regional Meteorological Aridity and Drought Characteristics of Baluchistan Province, Pakistan” used the already well-established methods for estimating characteristics of drought and aridity. The introduction section of the manuscript is not written well and there are a lot of confusing statements which make it very difficult to understand. Some of the issues are mentioned under the Specific Comments at the end of this document. The overall text of the manuscript is very confusing and not clear for understanding. I think the authors wrote the manuscript in a rush. The Discussion section of the manuscript discussed the results and therefore it is not look like a scientific discussion. I will suggest the authors to take their time and improve the overall manuscript.

Specific Comments:

1. Page 1, L 24-25: This statement “The temperature and precipitation data were obtained from the Pakistan Meteorological Department” make no sense abstract.

2. Page 1, L 29-30: This statement “The linear regression was used to find out the increasing aridity is increasing in the region” is confusing. Rephrase this for easy understanding.

3. Page 1, L 30-31: The statement “The result revealed that most stations are arid to semi-arid, and the highest increasing aridity is noted in Kalat (0.0065/annum)” can be like “The result revealed that most of the stations are arid to semi-arid, and the highest increasing aridity is noted in Kalat (0.0065/annum)”. This is one of the example of grammatical issues. Please check your manuscript thoroughly for such type of issues. I suggest to edit your manuscript from native English speaker before publication.

4. Keywords: A keyword “Aridity index AI” is mentioned in the list of keywords. It is not clear. Please mentioned clear keyword instead of confused keywords.

5. Page 2, L 56-58: The statements “The severe effects of drought are the negative effects on society and agriculture [8]. According Ofipcc (2013) defined drought as the time of unusually dry weather long enough to create a severe hydrological imbalance” are not clear. In the first statement you are talking about negative effects. What are these negative effects. The second statement is confusing in the start and not understandable.

6. Page 3, L 71-72: The statement “The Asian counties …”. I think you mean countries. Check your manuscript thoroughly for such type of issues.

7. Page 3, L 76-78: The statement “Less than 250 mm of rain falls annually throughout the entire province of Sindh, the majority of Balochistan, large portions of the Punjab, and the central regions of the Northern Areas [16]” is not clear. Rephrase it and make it clearer for easy understanding.

8. Page 8, L 178: Eq. (6) is not clear. Please check all equations of your manuscript thoroughly.

6. PLOS authors have the option to publish the peer review history of their article (what does this mean?). If published, this will include your full peer review and any attached files.

Reviewer #1: **Yes: **Muhammad Farhan Ul Moazzam

Reviewer #2: No

---

## [Author Response · Author response to Decision Letter 0]

18 Jul 2023

Ref. ESPR-D-21-18794R1: Estimation of regional meteorological aridity and drought characteristics of Baluchistan province, Pakistan

Point to Point Response to the Reviewer Comments

We are thankful to the editor and team for their timely feedback. We are also thankful to reviewer for their deep and valuable comments that improved the quality of the manuscript. We are submitting the point-by-point response to the comments and an updated manuscript file with highlighted revision as suggested by worthy reviewers. We are hopeful that the revised manuscript has improved to the level of satisfaction and approval. We also hope that now the manuscript meets the journal’s publication requirements. 

Editor

1. Thank you for stating the following financial disclosure:

This study is supported by National Natural Science Foundation of China (Grant No. 41877433) and the Hebei Natural Science Foundation and Key Basic Research (Grant No. 18963301D).

Author Response: The funders had no role in study design, data collection and analysis, decision to publish, or preparation of the manuscript.

2. We note that Figures 1,2, 5 and 6 in your submission contain [map/satellite] images which may be copyrighted. All PLOS content is published under the Creative Commons Attribution License (CC BY 4.0), which means that the manuscript, images, and Supporting Information files will be freely available online, and any third party is permitted to access, download, copy, distribute, and use these materials in any way, even commercially, with proper attribution. For these reasons, we cannot publish previously copyrighted maps or satellite images created using proprietary data, such as Google software (Google Maps, Street View, and Earth). For more information, see our copyright guidelines: http://journals.plos.org/plosone/s/licenses-and-copyright.

Author Response: The figures 1, 2, 5 and 6 in our submissions are the author own work and further a copy right holder letter uploaded to fulfil the journal requirements. The figures are modified after Rafiq M, Li YC, Cheng Y, Rahman G, Ali A, et al. (2022) Spatio-statistical analysis of temperature and trend detection in Baluchistan, Pakistan. Ecological Questions 33: 67-78. The research paper is published in under a Creative Commons Attribution License and the authors have right to publish or reprint it.

Author Response: No coding used in this manuscript. 

4. We note that the grant information you provided in the ‘Funding Information’ and ‘Financial Disclosure’ sections do not match. Please see below for your reference.

Author Response: This study is supported by National Natural Science Foundation of China (Grant No. 41877433) and the Hebei Natural Science Foundation and Key Basic Research (Grant No. 18963301D).

Also added in the Cover Letter

Author Response: Updated and also added in the cover Letter as “The authors have declared that no competing interests exist.”

6. We note that Figures 1,2, 5 and 6 in your submission contain [map/satellite] images which may be copyrighted. All PLOS content is published under the Creative Commons Attribution License (CC BY 4.0), which means that the manuscript, images, and Supporting Information files will be freely available online, and any third party is permitted to access, download, copy, distribute, and use these materials in any way, even commercially, with proper attribution. For these reasons, we cannot publish previously copyrighted maps or satellite images created using proprietary data, such as Google software (Google Maps, Street View, and Earth). For more information, see our copyright guidelines: http://journals.plos.org/plosone/s/licenses-and-copyright.

Author Response: The Content Permission form uploaded. These figures are prepared by one of the author and there is no restriction on its distribution and reprinting or publishing.

7. Please ensure that you refer to Table 2 and Table 5 in your text as, if accepted, production will need this reference to link the reader to the Table.

Author Response: Reference to Table 2 and Table 5 added in the revised manuscript.

Reviewer I

1. The paper is well organized and well written according to the journal format. English language is good however, I have following concerns before accepting this manuscript. 

Author Response: Thank you for your positive feedback. We revised the manuscript carefully.

2. Dear authors, I found some grammatical and technical issues in the abstract section therefore I will suggest to rewrite and align it with international journal standards. 

Author Response: We revised the abstract and tried to remove all the grammatical and technical issues. Most of the portion is rewritten to meet the international journal standard.

3. The given keywords are only three in numbers and there should be at least five keywords. 

Author Response: More keywords are added in the revised manuscript.

4. I will suggest to improve the writing of the introduction section and also include some further latest and updated literature regarding drought and aridity in the region. 

Author Response: The introduction section has been improved with more latest and updated literature.

5. In the study area section “From a morphological perspective, the study area (Baluchistan province) is a mountainous, arid in southwest territory of Pakistan (Fig .1).” morphology is not an accurate word. Replace it with some more suitable word. 

Author Response: The study area section has been revised and the inappropriate words have been removed in the revised section.

6. Correct the given latitude and longitude in the study area section “The region is located between 30.12 oN and 67.01 oE latitude and longitude, respectively.” 

Author Response: The latitude and longitude in the study area section has been revised.

7. There is a lot of grammatical and technical mistakes in the study area section like (i) “and are covered in dry terrain, are west of the Baluchistan plains.” (ii) “The bulk of the province is covered by the Kharan desert in the southwest” (iii) “Additionally, the province is crucial geographically for both regional trade and security.” (iv) “Baluchistan makes up 4% of Pakistan's overall land mass” (v) “Due to terrain brought on by monsoon and western storms, there are variations in rainfall and temperature dispersion.” (vi) “Baluchistan primarily experiences erratic weather that ranges from 30 mm to 397 mm annually”.

Author Response: Thank you worthy reviewer for pointing out these errors and mistakes in our manuscript. We carefully revised the manuscript and removed all these issues. We hope that the revised manuscript will be according to the language structure and all technical errors have been removed.

8. Figure 1 needs to be corrected as it does not reflect the actual topography of the study area. 

Author Response: Figure 1 has been revised

9. Rewrite the first sentence of the data collection and method section. 

Author Response: We revised this section

10. Rewrite the caption of table 1. Rewrite the caption of table 2. Rewrite table 3 caption. Rewrite the caption of table 4. Rewrite the caption of table 5. Rewrite the caption of Figure 4. Rewrite the caption of Figure 5.

Author Response: We revised all captions of the tables and figures

11. If this manuscript is based on SPEI then why the authors are focusing on SPI in the methodology section.

Author Response: We revised this section of the manuscript

12. The discussion section needs to be carefully studied and revised in light of more scientific literature with analytical reasoning. 

Author Response: We carefully rewrite the discussion section and relate the results with latest and updated literature with more analytical reasoning.

13. The conclusion section needs proofreading for grammatical correction 

Author Response: We revised the Conclusion section very carefully to remove the grammatical errors.

Reviewer II

1. The work in the manuscript titled “Estimation of Regional Meteorological Aridity and Drought Characteristics of Baluchistan Province, Pakistan” used the already well-established methods for estimating characteristics of drought and aridity. The introduction section of the manuscript is not written well and there are a lot of confusing statements which make it very difficult to understand. Some of the issues are mentioned under the Specific Comments at the end of this document. The overall text of the manuscript is very confusing and not clear for understanding. I think the authors wrote the manuscript in a rush. The Discussion section of the manuscript discussed the results and therefore it is not look like a scientific discussion. I will suggest the authors to take their time and improve the overall manuscript.

Author Response: Thank you respected reviewer for your encouraging words. We revised the whole manuscript in light of the worthy reviewer comments and hopefully now it will be in much improved form.

2. Page 1, L 24-25: This statement “The temperature and precipitation data were obtained from the Pakistan Meteorological Department” make no sense abstract. 

Author Response: Removed this sentence from the revised manuscript.

3. Page 1, L 29-30: This statement “The linear regression was used to find out the increasing aridity is increasing in the region” is confusing. Rephrase this for easy understanding. 

Author Response: We rephrased all such confusing sentences in the revised manuscript. Hopefully you will find it in much better form.

4. Page 1, L 30-31: The statement “The result revealed that most stations are arid to semi-arid, and the highest increasing aridity is noted in Kalat (0.0065/annum)” can be like “The result revealed that most of the stations are arid to semi-arid, and the highest increasing aridity is noted in Kalat (0.0065/annum)”. This is one of the example of grammatical issues. Please check your manuscript thoroughly for such type of issues. I suggest to edit your manuscript from native English speaker before publication. 

Author Response: It has been revised to clarify the results and statement in the manuscript and avoid any confusing statements.

5. Keywords: A keyword “Aridity index AI” is mentioned in the list of keywords. It is not clear. Please mentioned clear keyword instead of confused keywords. 

Author Response: Keywords have been revised. Hopefully it will be according to your suggestions.

6. Page 2, L 56-58: The statements “The severe effects of drought are the negative effects on society and agriculture [8]. According Ofipcc (2013) defined drought as the time of unusually dry weather long enough to create a severe hydrological imbalance” are not clear. In the first statement you are talking about negative effects. What are these negative effects. The second statement is confusing in the start and not understandable. 

Author Response: We revised all these sentences and added more scientific literature. Removed all the grammatical and technical errors from the manuscript.

7. Page 3, L 71-72: The statement “The Asian counties …”. I think you mean countries. Check your manuscript thoroughly for such type of issues. 

Author Response: Yes, respected reviewer the word is countries and corrected in the revised manuscript.

8. Page 3, L 76-78: The statement “Less than 250 mm of rain falls annually throughout the entire province of Sindh, the majority of Balochistan, large portions of the Punjab, and the central regions of the Northern Areas [16]” is not clear. Rephrase it and make it clearer for easy understanding. 

Author Response: We revised this section of the manuscript. “Most of the Baluchistan, Sindh and the southern and central parts of Punjab receives less than 250 mm of rainfall annually”.

9. Page 8, L 178: Eq. (6) is not clear. Please check all equations of your manuscript thoroughly. 

Author Response: It is checked and revised

---

## [Decision Letter · Decision Letter 1]

23 Aug 2023

PONE-D-23-15469R1Estimation of regional meteorological aridity and drought characteristics in Baluchistan province, PakistanPLOS ONE

Dear Dr. Rahman,

Thank you for submitting your manuscript to PLOS ONE. After careful consideration, we feel that it has merit but does not fully meet PLOS ONE’s publication criteria as it currently stands. Therefore, we invite you to submit a revised version of the manuscript that addresses the points raised during the review process.

We look forward to receiving your revised manuscript.

Kind regards,

Salim Heddam

Academic Editor

PLOS ONE

Additional Editor Comments:

Reviewer 1. The authors have addressed all my comments. Therefore, I accept this manuscript to be published in PLOS One Journal.

Reviewer 3. This paper reports the Estimation of Regional Meteorological Aridity and Drought Characteristics of Baluchistan Province, Pakistan. The aim of this study was to examine extreme climatic conditions such as drought in the province of Baluchistan, using ten meteorological stations. Overall, the themes are interesting as they show the climatic situation in this province in order to prepare the government to take adaptation and mitigation decisions to counter the effects of climate change on local populations. I suggest Major revision.

My main concerns:

1. I noticed that there is no regionalization analysis in this study, despite the reference in the title: "Estimation of regional meteorological aridity and drought characteristics in Baluchistan province, Pakistan". Therefore, a regionalization study using multivariate statistics such as PCA, Ward clustering with Ecludian distance, k-means or Moran's spatial autocorrelation...etc., should be added.

2. Numerous studies on drought in Pakistan have been reported in the literature, but the authors do not sufficiently summarize them in the introduction, nor do they show how the study carried out is new compared to those that preceded it, which are, for example, the following:

- Ahmed, K., Shahid, S., &Nawaz, N. (2018). Impacts of climate variability and change on seasonal droughtcharacteristics of Pakistan. Atmospheric Research. doi:10.1016/j.atmosres.2018.08.020

- Jamro, Shoaib, FalakNazChanna, Ghulam Hussain Dars, Kamran Ansari, and Nir Y. Krakauer. 2020. "Exploring the Evolution of DroughtCharacteristics in Balochistan, Pakistan" Applied Sciences 10, no. 3: 913. https://doi.org/10.3390/app10030913

-Jamro, Shoaib, Ghulam Hussain Dars, Kamran Ansari, and Nir Y. Krakauer. 2019. "Spatio-Temporal Variability of Drought in Pakistan Using Standardized Precipitation Evapotranspiration Index" Applied Sciences 9, no. 21: 4588. https://doi.org/10.3390/app9214588

-Dilawar, Adil, Baozhang Chen, Arfan Arshad, Lifeng Guo, Muhammad Irfan Ehsan, Yawar Hussain, Alphonse Kayiranga, Simon Measho, Huifang Zhang, Fei Wang, and et al. 2021. "TowardsUnderstanding Variability in Droughts in Response to Extreme Climate Conditions over the Different Agro-Ecological Zones of Pakistan" Sustainability 13, no. 12: 6910. https://doi.org/10.3390/su13126910

3. In this manuscript, neither drought characteristics such as drought duration or intensity, nor the maximum interannual variation of the SPEI or the aridity index have been carried out according to the meteorological stations of the present contribution.

4. As with other drought indices, a long reference period (30-50+ years) is required to increase the reliability of the SPEI drought estimate. However, the Barkhan station does not meet this condition (2009-2021). Drought analyses for this station should be taken with great caution or excluded from the study.

5. All the results of your analyses are not compared with those found in the literature.

6. Why haven't you carried out a quality control study on your data, for example to homogenize precipitation data, identify temperature extremes, etc.? It would be necessary to add descriptive statistics to your data before calculating the SPI and the aridity index.

7. In the discussion section, the authors do not show how the aridity index differs from the SPEI index in your analyses.

8. Finally, the SPEI can be calculated on a time scale ranging from 1 to 48 months. I'd like to know why you prefer SPEI at 1 and 6 months.

Minor but important issue:

1. In the summary, line 29-30, please rewrite the sentence.

2. In the summary, line 31, replace (0.0065/year) with year to clarify things for readers.

3. In the abstract, SPEI at 1 and 6 months, rewrite as SPEI at 1 and 6 months.

4. In the abstract, remove the hyphen before the expression aridity (Key words: - Aridity).

5. In the introduction, delete the second comma after line 59. The same applies to line 65.

6. Please quote a reference for the remark in line 71 of the introduction.

7. In the figure on line 111, please replace the unit of elevation with the meter, which is more fluid and easier to understand. Then, a more gradated coloring will be useful to highlight the differences in terrain leveling according to the geographical positions of the weather stations.

8. Please rewrite lines 115 to 117, as the text explanation is not fluent, and specify the type of data: daily or monthly.

9. Line 119: (Table.1), rewrite as (Table 1).

10 . Please specify the software used and its reference in your study.

11. On line 129, replace "the dot" with a "colon" after the reference and number the equation.

11 . On lines 145-147, a reference is required.

12 Lines 149 and 151, a reference is required.

13 Line 152, the equation is not the first but the second for numbering, please rectify for all others.

14 The equation on line 178, please review the question marks in the formula.

15 For each equation, a reference is required.

16 10. In figure 3, line 279, please add the respective p. values to the trend curves for the aridity and SPEI indices.

Reviewers' comments:

Reviewer's Responses to Questions

**Comments to the Author**

1. If the authors have adequately addressed your comments raised in a previous round of review and you feel that this manuscript is now acceptable for publication, you may indicate that here to bypass the “Comments to the Author” section, enter your conflict of interest statement in the “Confidential to Editor” section, and submit your "Accept" recommendation.

Reviewer #1: All comments have been addressed

Reviewer #3: (No Response)

2. Is the manuscript technically sound, and do the data support the conclusions?

Reviewer #1: Yes

Reviewer #3: Yes

3. Has the statistical analysis been performed appropriately and rigorously? 

Reviewer #1: Yes

Reviewer #3: No

4. Have the authors made all data underlying the findings in their manuscript fully available?

Reviewer #1: Yes

Reviewer #3: Yes

5. Is the manuscript presented in an intelligible fashion and written in standard English?

Reviewer #1: Yes

Reviewer #3: Yes

6. Review Comments to the Author

Reviewer #1: (No Response)

Reviewer #3: This paper reports the Estimation of Regional Meteorological Aridity and Drought Characteristics of Baluchistan Province, Pakistan. The aim of this study was to examine extreme climatic conditions such as drought in the province of Baluchistan, using ten meteorological stations. Overall, the themes are interesting as they show the climatic situation in this province in order to prepare the government to take adaptation and mitigation decisions to counter the effects of climate change on local populations. I suggest Major revision.

My main concerns:

1. I noticed that there is no regionalization analysis in this study, despite the reference in the title: "Estimation of regional meteorological aridity and drought characteristics in Baluchistan province, Pakistan". Therefore, a regionalization study using multivariate statistics such as PCA, Ward clustering with Ecludian distance, k-means or Moran's spatial autocorrelation...etc., should be added.

2. Numerous studies on drought in Pakistan have been reported in the literature, but the authors do not sufficiently summarize them in the introduction, nor do they show how the study carried out is new compared to those that preceded it, which are, for example, the following:

- Ahmed, K., Shahid, S., &Nawaz, N. (2018). Impacts of climate variability and change on seasonal droughtcharacteristics of Pakistan. Atmospheric Research. doi:10.1016/j.atmosres.2018.08.020

- Jamro, Shoaib, FalakNazChanna, Ghulam Hussain Dars, Kamran Ansari, and Nir Y. Krakauer. 2020. "Exploring the Evolution of DroughtCharacteristics in Balochistan, Pakistan" Applied Sciences 10, no. 3: 913. https://doi.org/10.3390/app10030913

-Jamro, Shoaib, Ghulam Hussain Dars, Kamran Ansari, and Nir Y. Krakauer. 2019. "Spatio-Temporal Variability of Drought in Pakistan Using Standardized Precipitation Evapotranspiration Index" Applied Sciences 9, no. 21: 4588. https://doi.org/10.3390/app9214588

-Dilawar, Adil, Baozhang Chen, Arfan Arshad, Lifeng Guo, Muhammad Irfan Ehsan, Yawar Hussain, Alphonse Kayiranga, Simon Measho, Huifang Zhang, Fei Wang, and et al. 2021. "TowardsUnderstanding Variability in Droughts in Response to Extreme Climate Conditions over the Different Agro-Ecological Zones of Pakistan" Sustainability 13, no. 12: 6910. https://doi.org/10.3390/su13126910

3. In this manuscript, neither drought characteristics such as drought duration or intensity, nor the maximum interannual variation of the SPEI or the aridity index have been carried out according to the meteorological stations of the present contribution.

4. As with other drought indices, a long reference period (30-50+ years) is required to increase the reliability of the SPEI drought estimate. However, the Barkhan station does not meet this condition (2009-2021). Drought analyses for this station should be taken with great caution or excluded from the study.

5. All the results of your analyses are not compared with those found in the literature.

6. Why haven't you carried out a quality control study on your data, for example to homogenize precipitation data, identify temperature extremes, etc.? It would be necessary to add descriptive statistics to your data before calculating the SPI and the aridity index.

7. In the discussion section, the authors do not show how the aridity index differs from the SPEI index in your analyses.

8. Finally, the SPEI can be calculated on a time scale ranging from 1 to 48 months. I'd like to know why you prefer SPEI at 1 and 6 months.

Minor but important issue:

1. In the summary, line 29-30, please rewrite the sentence.

2. In the summary, line 31, replace (0.0065/year) with year to clarify things for readers.

3. In the abstract, SPEI at 1 and 6 months, rewrite as SPEI at 1 and 6 months.

4. In the abstract, remove the hyphen before the expression aridity (Key words: - Aridity).

5. In the introduction, delete the second comma after line 59. The same applies to line 65.

6. Please quote a reference for the remark in line 71 of the introduction.

7. In the figure on line 111, please replace the unit of elevation with the meter, which is more fluid and easier to understand. Then, a more gradated coloring will be useful to highlight the differences in terrain leveling according to the geographical positions of the weather stations.

8. Please rewrite lines 115 to 117, as the text explanation is not fluent, and specify the type of data: daily or monthly.

9. Line 119: (Table.1), rewrite as (Table 1).

10 . Please specify the software used and its reference in your study.

11. On line 129, replace "the dot" with a "colon" after the reference and number the equation.

11 . On lines 145-147, a reference is required.

12 Lines 149 and 151, a reference is required.

13 Line 152, the equation is not the first but the second for numbering, please rectify for all others.

14 The equation on line 178, please review the question marks in the formula.

15 For each equation, a reference is required.

16 10. In figure 3, line 279, please add the respective p. values to the trend curves for the aridity and SPEI indices.

7. PLOS authors have the option to publish the peer review history of their article (what does this mean?). If published, this will include your full peer review and any attached files.

Reviewer #1: **Yes: **Muhammad Farhan Ul Moazzam

Reviewer #3: No

---

## [Author Response · Author response to Decision Letter 1]

30 Sep 2023

We are thankful to the editor and team for their timely feedback. We are also thankful to reviewer for their deep and valuable comments that improved the quality of the manuscript. We are submitting the point-by-point response to the comments and an updated manuscript file with highlighted revision as suggested by worthy reviewers. We are hopeful that the revised manuscript has improved to the level of satisfaction and approval. We also hope that now the manuscript meets the journal’s publication requirements. 

Editor

1. Thank you for stating the following financial disclosure:

This study is supported by National Natural Science Foundation of China (Grant No. 41877433) and the Hebei Natural Science Foundation and Key Basic Research (Grant No. 18963301D).

Reviewer I

1. Reviewer 1. The authors have addressed all my comments. Therefore, I accept this manuscript to be published in PLOS One Journal. 

Author Response: Thank you for your positive feedback and accepting our manuscript for publication in this prestigious journal. 

Reviewer 3

This paper reports the Estimation of Regional Meteorological Aridity and Drought Characteristics of Baluchistan Province, Pakistan. The aim of this study was to examine extreme climatic conditions such as drought in the province of Baluchistan. Overall, the themes are interesting as they show the climatic situation in this province in order to prepare the government to take adaptation and mitigation decisions to counter the effects of climate change on local populations. I suggest Major revision.

Author Response: Thank you respected reviewer for your encouraging words. We revised the whole manuscript in light of the worthy reviewer comments and hopefully now it will be in much improved form. Indeed it will be helpful in devising policies to combat drought in this arid zone of Pakistan.

1. noticed that there is no regionalization analysis in this study, despite the reference in the title: "Estimation of regional meteorological aridity and drought characteristics in Baluchistan province, Pakistan". Therefore, a regionalization study using multivariate statistics such as PCA, Ward clustering with Ecludian distance, k-means or Moran's spatial autocorrelation...etc., should be added. 

Author Response: We tried our best in the revision of this manuscript. We added further analysis as you suggested but unfortunately the number of meteorological stations are scarce in this region so this type of regionalization was not possible. 

2. Numerous studies on drought in Pakistan have been reported in the literature, but the authors do not sufficiently summarize them in the introduction, nor do they show how the study carried out is new compared to those that preceded it, which are, for example, the following:

- Ahmed, K., Shahid, S., &Nawaz, N. (2018). Impacts of climate variability and change on seasonal droughtcharacteristics of Pakistan. Atmospheric Research. doi:10.1016/j.atmosres.2018.08.020

- Jamro, Shoaib, FalakNazChanna, Ghulam Hussain Dars, Kamran Ansari, and Nir Y. Krakauer. 2020. "Exploring the Evolution of DroughtCharacteristics in Balochistan, Pakistan" Applied Sciences 10, no. 3: 913. https://doi.org/10.3390/app10030913

-Jamro, Shoaib, Ghulam Hussain Dars, Kamran Ansari, and Nir Y. Krakauer. 2019. "Spatio-Temporal Variability of Drought in Pakistan Using Standardized Precipitation Evapotranspiration Index" Applied Sciences 9, no. 21: 4588. https://doi.org/10.3390/app9214588

-Dilawar, Adil, Baozhang Chen, Arfan Arshad, Lifeng Guo, Muhammad Irfan Ehsan, Yawar Hussain, Alphonse Kayiranga, Simon Measho, Huifang Zhang, Fei Wang, and et al. 2021. "TowardsUnderstanding Variability in Droughts in Response to Extreme Climate Conditions over the Different Agro-Ecological Zones of Pakistan" Sustainability 13, no. 12: 6910. https://doi.org/10.3390/su13126910

Author Response: We added the suggested literature and summarize them in the Introduction section. We also highlighted how this study is different from the already conducted research studies in this region.

3. In this manuscript, neither drought characteristics such as drought duration or intensity, nor the maximum interannual variation of the SPEI or the aridity index have been carried out according to the meteorological stations of the present contribution.

Author Response: It has been revised to and drought characteristics like drought duration, intensity and severity results have been added in the revised manuscript. 

4. As with other drought indices, a long reference period (30-50+ years) is required to increase the reliability of the SPEI drought estimate. However, the Barkhan station does not meet this condition (2009-2021). Drought analyses for this station should be taken with great caution or excluded from the study.

Author Response: The results of Barkhan has been removed from the manuscript as suggested by worthy reviewer. 

5. All the results of your analyses are not compared with those found in the literature. 

Author Response: We revised all discussion section and tried to incorporate further discussion in light of previous literature.

6. Why haven't you carried out a quality control study on your data, for example to homogenize precipitation data, identify temperature extremes, etc.? It would be necessary to add descriptive statistics to your data before calculating the SPI and the aridity index.

Author Response: Descriptive statistics have been added in the revised manuscript.

7. In the discussion section, the authors do not show how the aridity index differs from the SPEI index in your analyses.

Author Response: We tried to differentiate the Aridity Index and SPEI in the revised manuscript.

8. Finally, the SPEI can be calculated on a time scale ranging from 1 to 48 months. I'd like to know why you prefer SPEI at 1 and 6 months. 

Author Response: in this manuscript we mainly focused on short term drought events therefore we used 1 and 6-months SPEI in this manuscript. 

Minor but important issue:

1. In the summary, line 29-30, please rewrite the sentence.

Author Response: We revised the suggested sentences.

2. In the summary, line 31, replace (0.0065/year) with year to clarify things for readers.

Author Response: We revised it as suggested by the worthy reviewer.

3. In the abstract, SPEI at 1 and 6 months, rewrite as SPEI at 1 and 6 months.

Author Response: Rewritten as suggested by the worthy reviewer.

4. In the abstract, remove the hyphen before the expression aridity (Key words: - Aridity).

Author Response: Hyphen removed from the Keywords.

5. In the introduction, delete the second comma after line 59. The same applies to line 65.

Author Response: Comma Deleted

6. Please quote a reference for the remark in line 71 of the introduction.

Author Response: Reference added to the mentioned sentence.

7. In the figure on line 111, please replace the unit of elevation with the meter, which is more fluid and easier to understand. Then, a more gradated coloring will be useful to highlight the differences in terrain leveling according to the geographical positions of the weather stations.

Author Response: The figures are revised as per your valuable suggestions.

8. Please rewrite lines 115 to 117, as the text explanation is not fluent, and specify the type of data: daily or monthly.

Author Response: We revised it as suggested by the worthy reviewer.

8. Line 119: (Table.1), rewrite as (Table 1).

Author Response: Corrected as per given suggestions.

10 . Please specify the software used and its reference in your study.

Author Response: Software added and discussed.

11. On line 129, replace "the dot" with a "colon" after the reference and number the equation.

Author Response: dot removed and colon added after reference and number has been added to the equation.

11 . On lines 145-147, a reference is required.

Author Response: We added reference as suggested by the worthy reviewer.

12 Lines 149 and 151, a reference is required.

Author Response: We added reference as suggested by the worthy reviewer.

13 Line 152, the equation is not the first but the second for numbering, please rectify for all others.

Author Response: corrected.

14 The equation on line 178, please review the question marks in the formula.

Author Response: corrected.

15 For each equation, a reference is required. 

Author Response: Reference is given.

16 10. In figure 3, line 279, please add the respective p. values to the trend curves for the aridity and SPEI indices.

Author Response: P values are added in trend curve. Thank you once again for your time and valuable suggestion.

---

## [Decision Letter · Decision Letter 2]

4 Oct 2023

Estimation of regional meteorological aridity and drought characteristics in Baluchistan province, Pakistan

PONE-D-23-15469R2

Dear Dr. Rahman

We’re pleased to inform you that your manuscript has been judged scientifically suitable for publication and will be formally accepted for publication once it meets all outstanding technical requirements.

Kind regards,

Salim Heddam

Academic Editor

PLOS ONE

Additional Editor Comments (optional):

Reviewer 1:The authors have addressed all my comments. Therefore I accept this manuscript to be published in PLoS One Journal

Reviewer 2:The authors responded all my question and I think this revised version is suitable for publication

Reviewers' comments:

Reviewer's Responses to Questions

**Comments to the Author**

1. If the authors have adequately addressed your comments raised in a previous round of review and you feel that this manuscript is now acceptable for publication, you may indicate that here to bypass the “Comments to the Author” section, enter your conflict of interest statement in the “Confidential to Editor” section, and submit your "Accept" recommendation.

Reviewer #3: All comments have been addressed

2. Is the manuscript technically sound, and do the data support the conclusions?

Reviewer #3: Yes

3. Has the statistical analysis been performed appropriately and rigorously? 

Reviewer #3: Yes

4. Have the authors made all data underlying the findings in their manuscript fully available?

Reviewer #3: Yes

5. Is the manuscript presented in an intelligible fashion and written in standard English?

Reviewer #3: Yes

6. Review Comments to the Author

Reviewer #3: The authors responded all my question and I think this revised version is suitable for publication

7. PLOS authors have the option to publish the peer review history of their article (what does this mean?). If published, this will include your full peer review and any attached files.

Reviewer #3: No

---

## [Editor Report · Acceptance letter]

21 Nov 2023

PONE-D-23-15469R2 

Estimation of regional meteorological aridity and drought characteristics in Baluchistan province, Pakistan 

Dear Dr. Rahman:

I'm pleased to inform you that your manuscript has been deemed suitable for publication in PLOS ONE. Congratulations! Your manuscript is now with our production department. 

Kind regards, 

on behalf of

Dr. Salim Heddam 

Academic Editor

PLOS ONE